# High-order dynamic localization and tunable temporal cloaking in ac-electric-field driven synthetic lattices

Shulin Wang[1,2,6], Chengzhi Qin[1,2,6], Weiwei Liu [1,2,6], Bing Wang [1,2]✉, Feng Zhou[1,2], Han Ye[1,2], Lange Zhao[1,2], Jianji Dong[1,2], Xinliang Zhang[1,2], Stefano Longhi [3,4]✉ & Peixiang Lu[1,2,5]✉

Dynamic localization (DL) of photons, i.e., the light-motion cancellation effect arising from lattice's quasi-energy band collapse under a synthetic ac-electric-field, provides a powerful and alternative mechanism to Anderson localization for coherent light confinement. So far only low-order DLs, corresponding to weak ac-fields, have been demonstrated using curved-waveguide lattices where the waveguide's bending curvature plays the role of ac-field as required in original Dunlap-Kenkre model of DL. However, the inevitable bending losses pose a severe limitation for the observation of high-order DL. Here, we break the weak-field limitation by transferring lattice concepts from spatial to synthetic time dimensions using fiber-loop circuits and observe up to fifth-order DL. We find that high-order DLs possess superior localization and robustness against random noise over lower-order ones. As an exciting application, by judiciously combining low- and high-order DLs, we demonstrate a temporal cloaking scheme with flexible tunability both for cloak's window size and opening time. Our work pushes DL towards high-order regimes using synthetic-lattice schemes, which may find potential applications in robust signal transmission, protection, processing, and cloaking.

Application of an electric field in solids gives rise to a series of coherent transport effects for electrons, ranging from dc electric-field driven Bloch oscillations[1,2] and Landau-Zener tunneling[3,4] to ac electric-field driven dynamic localization (DL)[5–7]. DL refers to a wave-motion can-celation phenomenon of an electron wave packet subject to an ac electric field at some magic values of the amplitude-to-frequency ratio. Such a localization mechanism stems from the quasi-energy band collapse of the ac-driven lattice[6] and provides an efficient approach for coherent light confinement, as highly alternative to the well-known Anderson localization that relies on disordered or quasi-periodic potentials[8–10]. The simplest case of DL was introduced by Dunlap and

Kenkre more than three decades ago for an electron hopping on a tight-binding lattice with nearest-neighbor hopping driven by an har-monic electric field[5], where quasi-energy band collapse is achieved[6] as the field's amplitude-to-frequency ratio takes a series of Bessel func-tion's roots, which are termed as different orders of DLs. The curved optical waveguide array system has provided a powerful setting to realize the Dunlap-Kenkre model of DL for photons[11–15], as well as other DL regimes[16–18]. Likewise, DL can arise in time-modulated resonator arrays[19,20], with applications in optical switching, filtering, and beam reshaping. However, so far only the first and second orders of DLs have been experimentally achieved in the curved waveguide array setup[11,12],

[1]Wuhan National Laboratory for Optoelectronics and School of Physics, Huazhong University of Science and Technology, Wuhan 430074, China. [2]Optics Valley Laboratory, Wuhan, Hubei 430074, China. [3]Dipartimento di Fisica, Politecnico di Milano, Piazza Leonardo da Vinci 32, I-20133 Milano, Italy. [4]IFISC (UIB-CSIC), Instituto de Fisica Interdisciplinar y Sistemas Complejos, E-07122 Palma de Mallorca, Spain. [5]Hubei Key Laboratory of Optical Information and Pattern Recognition, Wuhan Institute of Technology, Wuhan 430205, China. [6]These authors contributed equally: Shulin Wang, Chengzhi Qin, Weiwei Liu. ✉e-mail: wangbing@hust.edu.cn; stefano.longhi@polimi.it; lupeixiang@hust.edu.cn

limited by the considerable bending losses in highly curved arrays. To push DLs to high-order regimes may be desirable to enhance wave-packet localization strength so as to increase light-matter interaction. Furthermore, this localization enhancement may also facilitate the robust transport of wave packets against external noise.

Recently, the concepts of synthetic dimensions have emerged as ideal platforms for exploring various light transport behaviors, such as in time[21–32], frequency[33–38], and orbital angular momentum[39–42] spaces. Benefit from their intrinsic conveniences of control with external modulations, numerous fundamental physical concepts that are difficult to demonstrate in spatial lattices have been realized in synthetic dimensions, ranging from parity-time symmetry[23], the topological invariant's measurement[41], topological band windings[36–38] to non-Hermitian skin effect[30], and topological phase transitions in Floquet quasi-crystals[32]. Of all synthetic-dimension lattices, one promising platform is the temporal mesh lattice which can be constructed by mapping conceptually from two coupled fiber loops[22–32]. Thanks to the convenient and flexible introduction of modulations within the fiber loops, effective gauge fields can be readily created in the lattices, which give rise to various intriguing light transport phenomena, such as Bloch oscillations[22,25], Berry-curvature induced anomalous transport[27], Anderson localization[26,31], topological phase transitions and Hofstadter butterfly[32]. However, it remains fully unclear whether such discrete-time photonic quantum walks can realize the Dunlap-Kenkre model of DL, thus providing a fertile setting to demonstrate high-order DL regimes that have been so far elusive.

In this work, we show that discrete-time photonic quantum walks in synthetic temporal mesh lattices can indeed realize the continuous Dunlap-Kenkre model of DL, and report on the observations of high-order DLs under artificial ac electric fields of strong amplitude. We show that the width of lattice's quasi-energy band structure is controlled by a synthetic ac field, which collapses as the field's amplitude is tuned to a series of Bessel function's roots, corresponding to various orders of DLs. In experiments, we observe up to fifth-order DL and demonstrate that higher-order DLs possess smaller mean-square displacements during propagation and stronger robustness against stochastic noises than lower-order ones. However, contrary to the continuous Dunlap-Kenkre model, in the discrete-time quantum walks band flattening is not found at very high modulation amplitudes, where delocalization is ubiquitous. By combining the less-localized first-order and highly-localized fifth-order DLs, we propose a fully-reconfigurable temporal cloaking scheme where the first- and fifth-order DLs contribute to the opening and closing of the temporal cloak. We demonstrate that both the width and opening time of cloak can be freely tuned to fit the protection requirements of temporal events. The study on higher-order DLs and tunable temporal cloaking may find applications in robust signal transmission, processing, and temporal waveform reshaping.

## Results

### Theoretical model of high-order DLs

A synthetic temporal lattice can be created by connecting two coupled fiber loops with a tiny length difference, as illustrated in the inset of Fig. 1a. The length difference can induce a relative time delay for pulse traveling in the two loops, forming a pulse train that can be mapped to a discretized temporal lattice[22–32]. The lattice site $n$ labels the transverse pulse position while the step $m$ represents the pulse circulation number in the two loops. After each circulation, the pulse in the long loop obtains a time delay, corresponding to the hopping from $n$ to $n+1$. While in the short loop it obtains a time advance, resulting in the hopping from $n$ to $n-1$ sites. To create an additional ac electric field within the temporal lattice, we can apply opposite phase modulations of $\pm\phi(m)$ in the two fiber loops. Here we consider the simplest case of sinusoidally varying

phase modulation $\phi(m) = \Delta\phi\cos(\omega m + \varphi)$, where $\Delta\phi$, $\omega$, and $\varphi$ represent the modulation amplitude, frequency, and initial phase, respectively. Then a pulse will acquire a phase shift of $-\phi(m)$ from site $n$ to $n-1$ and an opposite phase of $\phi(m)$ when travelling from $n$ to $n+1$. Such a direction-dependent phase factor is a photonic analogue of Peierls phase, which corresponds to an effective vector potential $A_{\text{eff}} = \phi(m)$ applied in the temporal lattice[34,43,44]. Note that the vector potential itself is harmonically oscillating in time, it can lead to an ac electric field for photons[33], i.e., $E_{\text{eff}}(m) = -dA_{\text{eff}}/dm = \omega\Delta\phi\sin(\omega m + \varphi)$.

Under the action of the ac electric field, the pulse evolution in the temporal lattice can be described by the following equation

$$\begin{cases} u_n^m = \left[\cos(\beta)u_{n+1}^{m-1} + i\sin(\beta)v_{n+1}^{m-1}\right]e^{-i\phi(m)} \\ v_n^m = \left[i\sin(\beta)u_{n-1}^{m-1} + \cos(\beta)v_{n-1}^{m-1}\right]e^{i\phi(m)} \end{cases}, \quad (1)$$

where $u_n^m$ and $v_n^m$ denote the pulse amplitudes in short and long loops at lattice site $n$ and time step $m$. The power splitting ratio of the directional coupler is defined as $\sin^2(\beta)/\cos^2(\beta)$, with $\beta \in [0, \pi/2]$. Consider the eigen Bloch mode supported by the temporal lattice

$$|\psi\rangle = \begin{pmatrix} u_n^m \\ v_n^m \end{pmatrix} = \begin{pmatrix} U \\ V \end{pmatrix} e^{iQn}e^{i\theta m}, \quad (2)$$

where $(U, V)^{\text{T}}$ denotes the eigenvector, $Q$ and $\theta$ are the transverse Bloch momentum and longitudinal propagation constant, respectively. Substituting Eq. (2) into Eq. (1), we can obtain the instantaneous band structure of the temporal lattice (see Supplementary Note 1)

$$\theta_\pm[Q(m)] = \mp\cos(\beta)\cos[Q(m)] \pm \frac{\pi}{2}, \quad (3)$$

where $Q(m) = Q - \phi(m)$ is the time-dependent Bloch momentum, "$\pm$" denote the upper and lower branches of band structure. As shown in the band structure of Fig. 1b, the application of an ac electric field can induce the periodic shifting of Bloch momentum within a region of range $2\Delta\phi$ in the Brillouin zone. To obtain the averaging effect, we should ensure the adiabaticity and keep the continuous-time approximation valid by choosing a slow modulation frequency $\omega = 2\pi/M$, where the integer $M$ is the modulation period. In this limit, the time-independent quasi-energy band structure can be obtained by performing the time averaging of instantaneous band structure over one driving period, i.e., (see Supplementary Note 2)

$$\langle\theta_\pm\rangle = \frac{1}{M}\int_0^M \theta_\pm(m)dm = \mp J_0(\Delta\phi)\cos(\beta)\cos(Q) \pm \frac{\pi}{2}. \quad (4)$$

where $J_0(\Delta\phi)$ is the zeroth-order Bessel function. Figure 1c shows the quasi-energy band structure versus different modulation amplitude $\Delta\phi$ and its projection onto the $\langle\theta\rangle$-$\Delta\phi$ plane. It shows that the effect of the ac electric field is to modify the bandwidth, i.e., the scope of quasi-energy band structure. Specifically, the band-width will collapse as $\Delta\phi$ is tuned to a series of zeros of $J_0$ function (labelled by the red lines and dots), at which different orders of DLs occur. The exact mapping of the discrete-time photonic quantum walks with the continuous Dunlap-Kenkre model of DL, obtained for a coupling angle $\beta$ close to $\pi/2$ and a slow modulation frequency, is presented in Supplementary Note 2. To see how the ac electric field controls the lattice evolution dynamics, we consider a Bloch-mode wave packet impinging on the lattice with initial Bloch momentum $Q$, the averaged group velocity can be derived from the quasi-energy band structure, which reads

$$\langle v_{g,\pm}\rangle = -\partial\langle\theta_\pm\rangle/\partial Q = \mp J_0(\Delta\phi)\cos(\beta)\sin(Q). \quad (5)$$

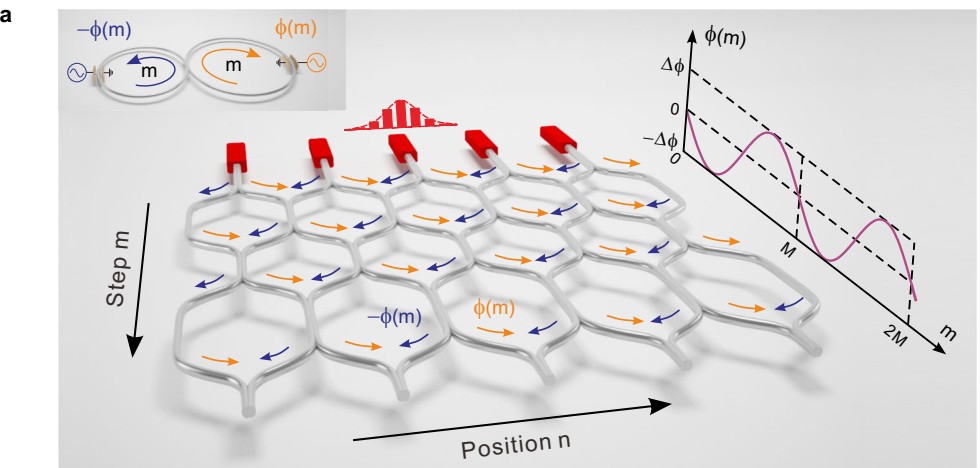

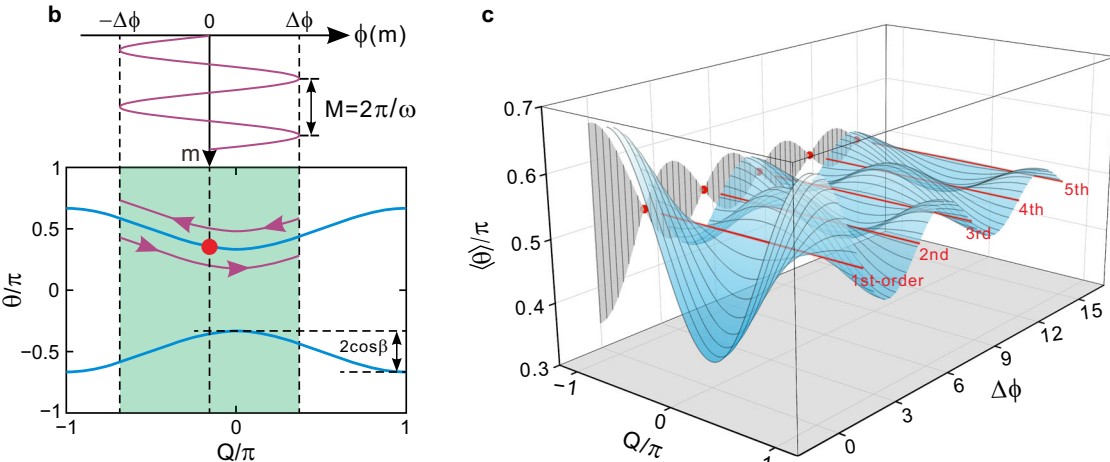

**Fig. 1 | Principle of different-order DLs in ac-driven synthetic temporal lattices.** **a** Schematic diagram of a synthetic temporal mesh lattice constructed by mapping from two coupled fiber loops as shown by the inset figure. The opposite sinusoidal phase modulations $\pm\phi(m)$ are incorporated into the two fiber loops to generate a direction-dependent phase factor accompanying light hopping in the temporal lattice, corresponding to an effective time-periodic vector potential and hence an ac electric field therein. **b** Instantaneous band structure of synthetic temporal lattice, where the Bloch momentum follows a periodic oscillation under the driving of ac electric field with driving period $M = 2\pi/\omega$ and driving amplitude $\Delta\phi$. **c** Quasi-energy band structure versus the driving amplitude $\Delta\phi$ and its projection onto the $\langle\theta\rangle$-$\Delta\phi$ plane, obtained by performing time averaging over the instantaneous band structure within one driving period $M$. Here only the upper band is plotted. The red lines denote the specific modulation amplitudes at $J_0$ Bessel function's zeros where the quasi-energy band structure collapses, corresponding to the occurrence of different-order DLs.

After a driving period $M$, the wave packet can accumulate a total transverse displacement

$$\Delta n_\pm = M\langle v_{g,\pm}\rangle = \mp MJ_0(\Delta\phi)\cos(\beta)\sin(Q). \tag{6}$$

Meanwhile, the packet will experience the envelope broadening during propagation, which can be quantitatively described by the averaged diffraction coefficient

$$\langle D_\pm\rangle = \partial^2\langle\theta_\pm\rangle/\partial^2 Q = \pm J_0(\Delta\phi)\cos(\beta)\cos(Q). \tag{7}$$

As $\Delta\phi$ takes one of the roots of $J_0$ function, both the wave packet transverse shifting and broadening vanish, $\langle v_{g,\pm}\rangle = \langle D_\pm\rangle = 0$, the packet will restore to its initial incident position with initial profile after each driving period, showing the characteristic features of periodic revival for DLs. However, since different orders of DLs share the common features of periodic wave-motion and broadening cancelation, one cannot distinguish them in terms of $\langle v_{g,\pm}\rangle$ and $\langle D_\pm\rangle$.

To distinguish different orders of DLs, one needs to inspect the explicit evolution process within a driving period. In fact, due to the

non-vanishing instantaneous group velocity $v_{g,\pm}(m) = -\partial\theta_\pm(m)/\partial Q = \mp\cos(\beta)\sin[Q - \Delta\phi\cos(\omega m + \varphi)]$ and diffraction coefficient $D_\pm(m) = \partial^2\theta_\pm(m)/\partial^2 Q = \pm\cos(\beta)\cos[Q - \Delta\phi\cos(\omega m + \varphi)]$, the wave packet experiences delocalization at each step within a driving period. Quantitatively, the degree of delocalization during propagation can be characterized by a statistic parameter of instantaneous mean-square displacement, $\langle n^2(m)\rangle$[5,11,31]

$$\langle n^2(m)\rangle = \frac{\sum_n[n^2(|u_n^m|^2 + |v_n^m|^2)]}{\sum_n(|u_n^m|^2 + |v_n^m|^2)}. \tag{8}$$

which measures the displacement of wave-packet with respect to the initial reference position. According to Eq. (8), $\langle n^2(m)\rangle$ depends quadratically on the occupied position $n$, indicating that it will increase with the packet's transverse displacement and diffraction spreading. Different-order DLs exhibit different wave-packet dynamics and hence $\langle n^2(m)\rangle$ evolutions. More specifically, the maximum mean-square displacement $\langle n^2(m)\rangle_{max}$ within a single driving period can be adopted to characterize the localization strength of different-order DLs.

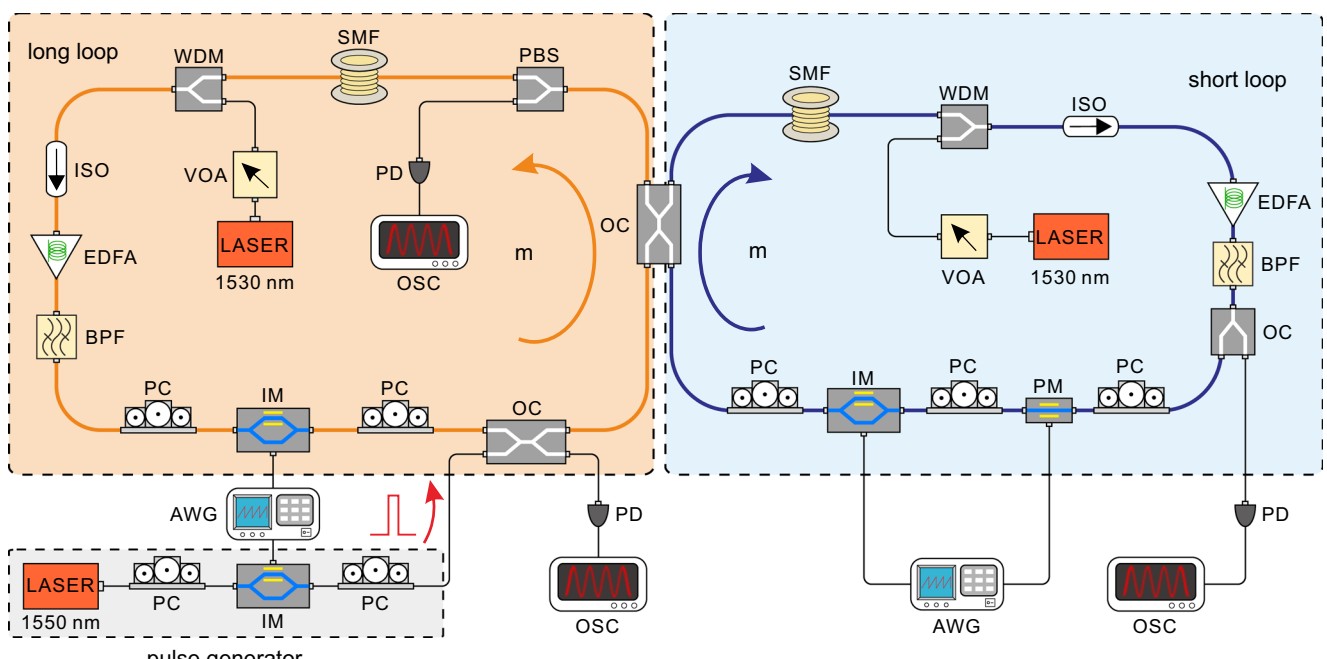

**Fig. 2 | Experimental setup.** The yellow and blue loops denote the long and short loops and the yellow and blue arrows represent the pulse circulation directions within the two loops. All optical and electric components are as follows: Polarization controller (PC), intensity modulator (IM), optical coupler (OC), arbitrary waveform generator (AWG), single mode fiber (SMF), polarization beam splitter (PBS), photodiode (PD), oscilloscope (OSC), variable optical attenuator (VOA), isolator (ISO), wavelength division multiplexer (WDM), erbium-doped fiber amplifier (EDFA), band-pass filter (BPF), phase modulator (PM).

## Experimental realization of high-order DLs

To verify the theoretical analysis, we experimentally build a coupled double fiber-loops circuit, as shown schematically in Fig. 2. Two fiber loops with an average length of ~5 km are connected via a 75:25 (corresponding to $\beta = \pi/3$) directional coupler. The length difference of the two loops is ~30 m, corresponding to a relative time delay of ~150 ns. The required sinusoidal phase modulation is provided by the incorporated phase modulator (PM) in the short loop driven by an arbitrary waveform generator (AWG). The initial Bloch-mode wave packet is prepared from a single optical pulse with a duration time of ~100 ns injected from the long fiber loop (see Materials and Methods). The detection of wave packet evolution at each step is realized by recording the pulse-train intensity distributions extracted from the two loops. Other details about the experimental setup and measurement are also provided in Materials and Methods.

In the experiment, we excite a Bloch-mode wave packet from the upper band as the incidence, which carries an initial Bloch momentum $Q = \pi/2$. Figure 3a depicts the packet transverse displacement $\Delta n$ after a single driving period as a function of the phase modulation amplitude $\Delta\phi$. Here the modulation frequency is fixed as $\omega = \pi/60$, corresponding to a driving period of $M = 2\pi/\omega = 120$. It shows that the displacement $\Delta n$ follows an oscillatory variation of $J_0$ Bessel function with the increase of $\Delta\phi$, which is in perfect accordance with theoretical prediction of Eq. (6). Specifically, $\Delta n = 0$ occurs at $\Delta\phi = 2.4$, 5.5, 8.7, 11.8, and 14.9, as denoted by the blue dots, clearly validating DLs from the first to fifth orders.

In Figs. 3b and 3c, we illustrate the simulated and experimental wave packet evolutions by choosing several specific modulation amplitudes of $\Delta\phi = 0$, 3.8, 2.4, 14.9. For comparison, we firstly consider the packet evolution in the absence of modulation with $\Delta\phi = 0$. In this case, the packet exhibits a constant group velocity of $v_g = -\cos(\beta)\sin(Q) = -0.5$, corresponding to a maximum left displacement of $\Delta n = v_g M = -60$, as shown in Fig. 3b(i). In Fig. 3c(i), we get a measured displacement of $\Delta n = -59.2$, which matches well with

the theoretical prediction. For a non-zero modulation amplitude $\Delta\phi = 3.8$, as shown in Figs. 3b(ii) and 3c(ii), the packet exhibits a maximum right displacement of $\Delta n = 24.5$, in accordance with theoretical result of $\Delta n = -MJ_0(\Delta\phi)\cos(\beta)\sin(Q) = 24.2$. It shows that the presence of modulation (ac electric field) can modify the wave-packet group velocity, which reaches the maximum displacement for the field-free case. Additionally, the wave packet manifests a curved evolution trajectory under the ac field driving with a time-varying group velocity, in contrast to the straight trajectory with a constant velocity for the field-free case. Also note that both the field-driven and field-free cases exhibit diffraction-free evolutions without packet broadening. This is attributed to the vanishing diffraction coefficient $\langle D \rangle = 0$ for $Q = \pi/2$ according to Eq. (7). Besides the rightward motion of wave packet, the ac driving can also lead to leftward motion of packet when the choice of modulation amplitude $\Delta\phi$ corresponds to a negative averaged group velocity.

We show the simulated and measured pulse intensity evolutions for the 1st- and 5th-order DLs in Figs. 3b(iii), 3b(iv), 3c(iii), and 3c(iv). One sees that the wave packet displays an oscillatory trajectory during propagation and restores to its initial position and profile after a single driving period. The packet for the 1st-order DL possesses much larger oscillation amplitude compared to that of the 5th-order one. Additionally, the wave packet experiences obvious diffraction spreading during the 1st-order DL process. In contrast, the packet width remains nearly unchanged with negligible broadening for the 5th-order DL. Hence, higher-order DLs possess much stronger localization strength than lower-order ones.

To quantitatively characterize the localization strength of different orders of DLs, we plot in Fig. 3d the measured mean-square displacement $\langle n^2(m) \rangle$ evolutions by choosing the 1st- and 5th-order cases. The comprehensive comparisons of pulse intensity evolutions and mean-square displacements from the 1st- to 5th-order DLs are provided in Supplementary Note 3. Here, at each step $m$, the mean-square displacement $\langle n^2(m) \rangle$ for the 5th-order DL is much smaller than that of the 1st-order one. As also summarized in the

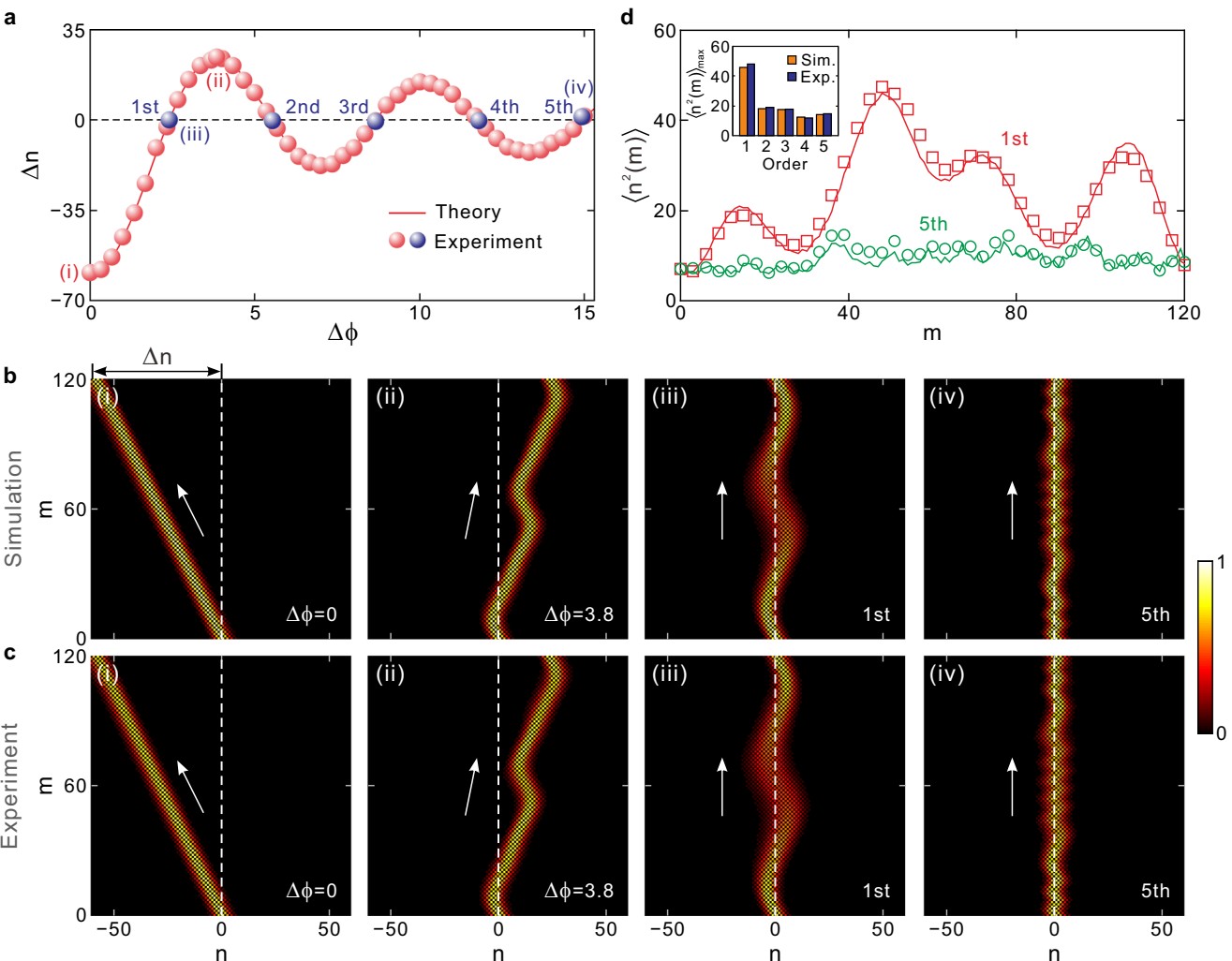

**Fig. 3 | Simulated and experimental results of different-order DLs. a** Transverse displacement of wave packet $\Delta n$ after a single driving period versus phase modulation amplitude $\Delta\phi$. **b** Simulated pulse intensity evolutions in one driving period with total step $M = 120$. (i) and (ii) correspond to directional transports with $\Delta\phi = 0$ and 3.8, respectively. (iii) and (iv) correspond to the 1st- and 5th-order DLs,

respectively. **c** Measured pulse intensity evolutions. (i)-(iv) correspond to the simulated results in **b**(i)-**b**(iv). **d** Mean-square displacement $\langle n^2(m) \rangle$ with respect to step $m$ for 1st- and 5th-order DLs. The inset figure shows the maximum mean-square displacement $\langle n^2(m) \rangle_{max}$ versus the order of DL.

inset of Fig. 3d, the maximum mean-square displacement $\langle n^2(m) \rangle_{max}$ decreases from 48.1 to 14.9 as the order increases from 1 to 5, clearly demonstrating that the wave packet can be better localized for the higher-order DL. Quantitatively, $\langle n^2(m) \rangle_{max}$ decreases by 61% from 1st to 2nd orders while it decreases by 31% and 21% from 2nd to 4th and 5th orders, respectively. As we discussed in Supplementary Note 5, the decrements of $\langle n^2(m) \rangle_{max}$ for higher-order DLs can be further improved by enlarging the driving period $M$. The mechanism for enhanced localization strength of high-order DL can be attributed to the faster variation rate of wave-packet momentum under larger ac electric field driving (see Supplementary Note 4). As shown schematically in Supplementary Fig. S4, a larger electric field can drive a Bloch wave packet to oscillate across a larger regime in the extended Brillouin zone, leading to more times of packet Bragg reflections at each Brillouin zone edge and center. More frequent Bragg reflections will cancel the net accumulated packet motion in one direction within a driving period and hence give rise to the stronger localization strength.

**Enhanced robustness of high-order DLs against random noises**
In previous sections, we have adopted the localization strength as a signature to distinguish different orders of DLs. In this section, we

will investigate the robustness of DLs against external random noises to further demonstrate the advantages of high-order DLs over low-order ones. Generally, we consider a randomly distributed, time-varying modulation-phase noise superimposed onto the sinusoidally-varying modulation phase, i.e., $\phi(m) = \Delta\phi\cos(\omega m + \varphi) + \phi_{noise}(m)$. To check the influence of different-type noises on the dynamics of DLs, we choose three representative noise models with respective uniform, Gaussian, and gamma probability density functions. Our simulation results reveal that different types of noises of comparable expected values and standard derivations give rise to nearly the same evolution dynamics, indicating that the enhanced robustness against noise is a universal behavior for high-order DLs that is applicable to a generally applied noise (see Supplementary Note 7 for detailed comparisons and discussions). In our experiment, we choose the simplest uniform noise as a representative example, where $\phi_{noise}(m)$ is randomly chosen from a uniform distribution over $[-\delta\phi/2, \delta\phi/2]$, with $\delta\phi$ being the range of noise. The superimposed modulation waveforms are schematically shown in Fig. 4a, where we have chosen both a strong and a weak ac field amplitude. In the presence of random noise, DLs of different orders will degrade, giving rise to the broadening of wave packet. Quantitatively, to describe the degree of wave-packet broadening,

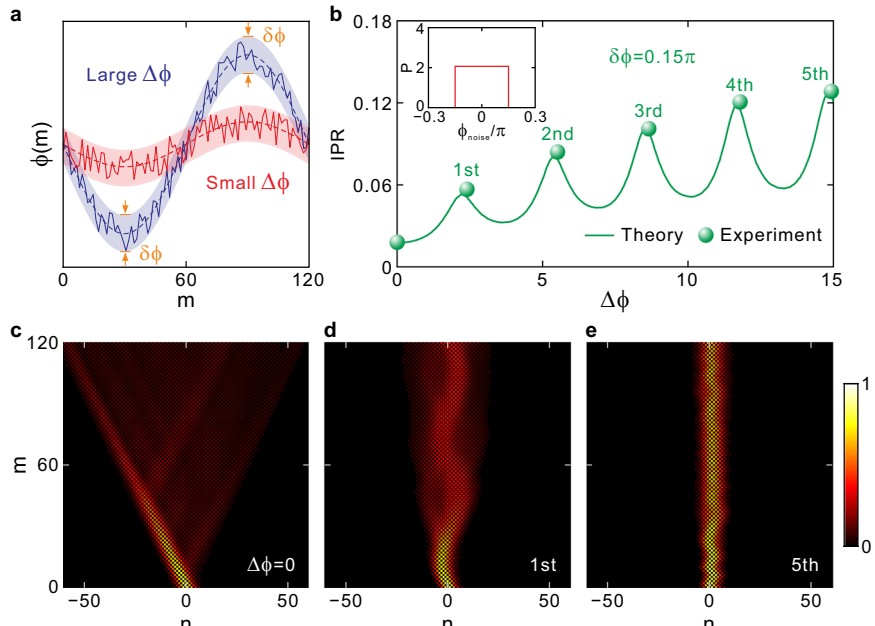

**Fig. 4 | Robustness comparison of different-order DLs against stochastic noises. a** Sinusoidal phase modulation with a superimposed random noise. **b** Inverse participation ratio IPR of wave packet after one modulation cycle at step $m = 120$ in the presence of external random noise. The inset depicts the probability density function $P(\phi_{noise})$ of uniform noise with $\delta\phi = 0.15\pi$. **c** Measured pulse intensity evolution for $\Delta\phi = 0$. **d, e** Measured pulse intensity evolutions for the 1st- and 5th-order DLs.

we utilize the inverse participation ratio (IPR), which is defined as[26,30]

$$IPR = \frac{\sum_n (|u_n^m|^2 + |v_n^m|^2)^2}{[\sum_n (|u_n^m|^2 + |v_n^m|^2)]^2}, \tag{9}$$

with $0 < IPR \leq 1$. A higher IPR reflects less wave packet spreading and thus better localization, also suggesting stronger robustness against external noise.

The solid curve in Fig. 4b depicts the theoretical variation of IPR after one driving period as the modulation amplitude increases continuously from $\Delta\phi = 0$ to $\Delta\phi = 15$. Here the probability density function of uniformly distributed noise is inserted in Fig. 4b, where we fix the uniform noise range at $\delta\phi = 0.15\pi$. It shows that the IPR reaches a peak value at each order of DLs, indicating that DL can efficiently suppress the noise-induced wave packet broadening. To verify this, in the experiment we choose modulation amplitudes at different orders of DLs, where the measured IPR can match well with the theoretical results. Moreover, as the order increases from 1 to 5, the IPR for different order DLs increases from 0.06 to 0.13, suggesting that higher-order DLs possess superior robustness against stochastic noises over lower-order ones. Figures 4c-4e illustrate the measured pulse intensity evolutions for $\Delta\phi = 0$, 2.4 (1st-order DL), and 14.9 (5th-order DL), respectively. Here for each case, we have performed statistic averaging for 10 times of experiment results. Without the ac electric field, the wave packet experiences the most serious expansion, showing cone-like (ballistic) packet boundaries (Fig. 4c). For the 1st-order DL, the wave packet width has expanded by about 5 times after one driving period (Fig. 4d). In contrast, the wave-packet width almost conserves for the 5th-order DL thanks to the stronger robustness against noise (Fig. 4e). More detailed pulse intensity evolutions from the 1st- to 5th-order DLs in the presence of the random noises are shown in Supplementary Note 6. The enhanced robustness of high-order DLs against noises

can be explained from the slope of quasi-energy band structures at each zero of $J_0$ Bessel function. As shown in Fig. 1c, the slope of $J_0$ Bessel function at each zero becomes smaller as the zero's order increases, such that under the same perturbation strength $\delta\phi$, higher-order DL at higher-order zero gets smaller bandwidth expansion from the collapsed point and hence stronger localization properties.

## Tunable temporal cloaking by combining higher- and lower-order DLs

As we demonstrate in previous sections, higher-order DL manifests smaller packet displacement and stronger robustness against noises, which are desirable for robust signal transmission. On the other hand, lower-order DL exhibits larger displacement within a driving period and can restore to its initial position after the driving period, which are helpful for signal delay and reconstruction. In this section, we combine both higher- and lower-order DLs to design a temporal cloaking scheme. As shown schematically in Fig. 5a, a wave packet impinges into the first region where we use fifth-order DL to realize robust signal transmission. At the interface of regions 1 and 2, we use first-order DL and introduce a constant relative phase shift to achieve wave-packet splitting. Such a splitting can circumvent an event, thus forming a temporal cloak. Then we switch back to high-order DL at the interface of regions 2 and 3, where packet recombination occurs after finishing one driving period.

The relative constant phase shift between regions 2 and 1 plays a role of an effective gauge potential, which can induce a constant band shifting and hence wave packet splitting. Assume that the constant biased phase in regions 1 and 2 is denoted by $\phi_l$, $(l = 1, 2)$, such that the band structures are $\theta_{l,\pm} = \mp\cos(\beta)\cos(Q - \phi_l)\pm\pi/2$ (Fig. 5b), with the corresponding eigen states given by

$$|\psi_{l,\pm}\rangle = \begin{pmatrix} U_{l,\pm} \\ V_{l,\pm} \end{pmatrix} e^{iQn} e^{i\theta_{l,\pm}m}, \tag{10}$$

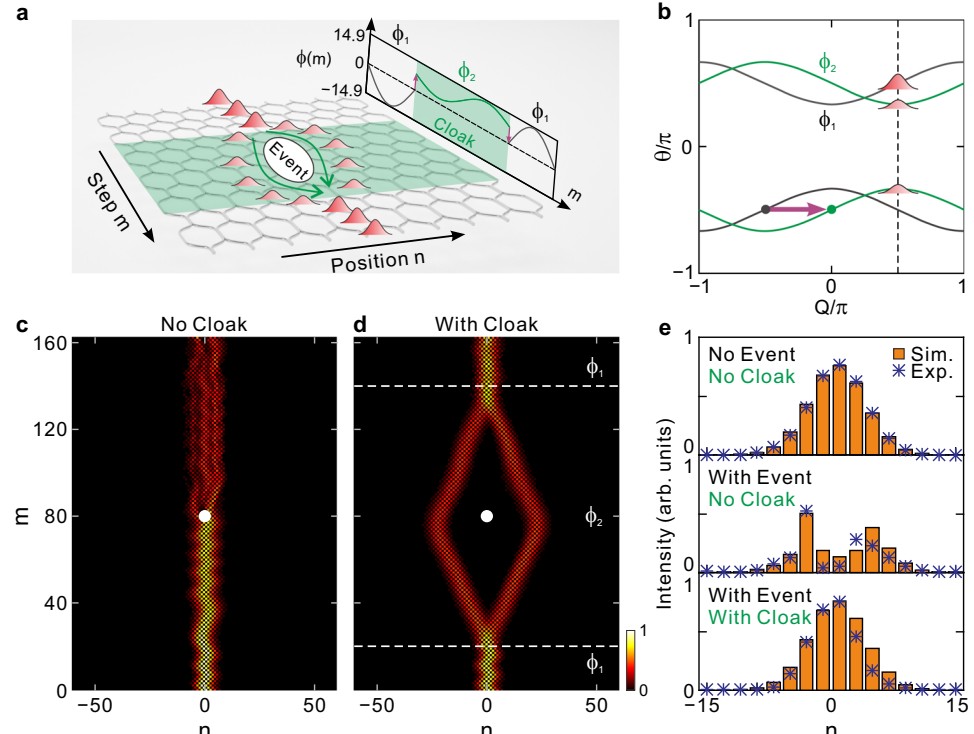

**Fig. 5 | Temporal cloaking with the combination of higher- and lower-order DLs. a** Schematic of the temporal cloaking, where the green zone with modulation phase $\phi_2$ denotes the cloak region and the gray zones with modulation phase $\phi_1$ represent the regions outside the cloak. **b** Relative band structure shift for the two regions caused by abrupt change of the modulation phase. **c, d** Pulse intensity evolutions without and with the temporal cloak. The white point denotes the temporal absorption event. **e** Pulse intensity distribution sliced at $m = 162$. The orange blocks and blue asterisks represent the simulated and experimental results, respectively.

where

$$\begin{pmatrix} U_{l,\pm} \\ V_{l,\pm} \end{pmatrix} = \frac{1}{\sqrt{1+e^{\mp 2\lambda_l}}} \begin{pmatrix} 1 \\ -e^{\mp \lambda_l} e^{-i(Q-\phi_l)} \end{pmatrix}, \quad (11)$$

and $\lambda_l = \mathrm{arsinh}[\cot(\beta)\sin(Q-\phi_l)]$. Consider a wave packet incidence from the upper band in region 1, it will exhibit refraction at the interface of region 1 and 2. According to the conservation law of Bloch momentum at the interface, as denoted by the dashed line in Fig. 5b, the incident wave packet in region 1 can match two packets of the two branches in region 2, i.e.,

$$\begin{pmatrix} U_{1,+} \\ V_{1,+} \end{pmatrix} = c_{2,+} \begin{pmatrix} U_{2,+} \\ V_{2,+} \end{pmatrix} + c_{2,-} \begin{pmatrix} U_{2,-} \\ V_{2,-} \end{pmatrix}, \quad (12)$$

where $c_{2,+}$ and $c_{2,-}$ are the occupation coefficients of the upper and lower bands. By combing Eqs. (11) and (12), we further obtain

$$\begin{cases} c_{2,+} = (V_{1,+}U_{2,-} - U_{1,+}V_{2,-})/(U_{2,-}V_{2,+} - U_{2,+}V_{2,-}) \\ c_{2,-} = (U_{1,+}V_{2,+} - V_{1,+}U_{2,+})/(U_{2,-}V_{2,+} - U_{2,+}V_{2,-}) \end{cases}. \quad (13)$$

Specifically, for $\phi_2 - \phi_1 = \pi/2$, we can achieve $|c_{2,+}|^2 = |c_{2,-}|^2 = 0.5$, indicating the wave packet is equally splitting in power. In region 2, the two split wave packets exhibit 1st-order DLs under a low-amplitude ac electric field driving. Since the two packets occupying two bands have opposite group velocities, they will transport separately with mirror-symmetric trajectories, therefore creating a time window, i.e., a temporal cloak. After finishing one-period DL in region 2, the two packets restore to their initial positions and manifest a packet recombination at the interface of regions 2 and 3, leading to the closing of the temporal cloak. Note that due to the

periodic revival nature of DL, the packet recombination process is just the time reversal of the packet splitting. In region 3, the packet then goes on exhibiting fifth-order DL.

In our experiments, the temporal event to be cloaked is mimicked by the pulse intensity absorption between $n = -1$ and $n = 1$ at $m = 80$ step realized by changing the intensity modulators' transmittances of two loops. For comparison, we first show the wave packet evolution of the 5th-order DL without a temporal cloak in Fig. 5c. It shows that even though the 5th-order DL manifests strong robustness against the noise in the driving electric field, it is still severely affected by the absorption event. In contrast, by introducing the above temporal cloak, as depicted in Fig. 5d, the absorption event can be circumvented perfectly by the time window, such that the packet restores to its input state of the 5th-order DL after the cloaking region. Figure 5e illustrates the output wave packet intensity distributions detected at step $m = 162$ with and without the temporal cloak. By comparing with the packet direct transmission without absorption event, the output packet is completely unaffected with the protection of the temporal cloak, as if the event does not exist.

Next, we demonstrate that the temporal cloak possesses flexible tunability both in the cloaking window size and its opening time. As shown in Fig. 6a, since the two split wave packets belonging to the lower and upper bands have positive and negative group velocities, they will propagate towards opposite directions. The width of the cloaking window is defined by the maximum spacing of the two split packets, which is reached at the center of one driving period $m = M/2$, i.e.,

$$W_{\max} = \int_0^{M/2} |v_{g,-}(m) - v_{g,+}(m)| dm = MW_0, \quad (14)$$

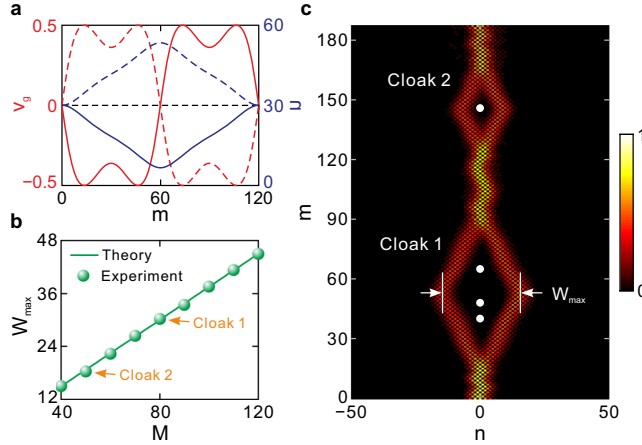

**Fig. 6 | Tunability of the cloak's window size and opening time. a** Transverse group velocity $v_g$ and displacement $\Delta n$ with respect to step $m$. The dashed and solid curves correspond to the lower and upper bands, respectively. **b** Cloak width $W_{max}$ varying with driving period $M$. **c** Measured pulse intensity evolution with two cloaks. For cloaks 1 and 2, the corresponding driving periods are $M = 80$ and $M = 50$, respectively.

where $W_0 = 2\cos(\beta) \int_0^{1/2} \sin[\Delta\phi \sin(2\pi m/M)] d(m/M)$, denoting the averaging width increase within one step (see Supplementary Note 8). Equation (14) suggests that the cloaking window width is proportional to the driving period $M$, as also plotted in Fig. 6b. For experimental demonstrations, we design two cloaks of different sizes by choosing two driving periods of $M = 80$ and $M = 50$, respectively. Figure 6c shows the packet evolution with the two cloaks. The width of the first cloak reaches $W_{max} = 31$ with $W_0 = 0.37$, such that it can circumvent three absorption events. While the size of the second cloak is smaller with the width being $W_{max} = 17.5$, within which only one absorption event is cloaked. With its flexible tunability, our temporal cloak can adapt to random perturbation events with different trigging and duration time.

It is worth comparing our temporal cloaking mechanism to the conventional dispersion-based temporal cloaking schemes. Previously, the temporal cloaking is achieved in optical fibers where the normal and anomalous group velocity dispersions (GVDs) of optical pulses contribute to the opening and closing of temporal cloak[45,46]. Nevertheless, it is technically challenging to modify and modulate the GVD coefficient of optical fibers, which highly limits the manipulation of cloak's window size and opening time. Here, by combing both lower- and higher-order DLs using strategic ac-field driving, we demonstrated a temporal cloak with dynamically controllable cloaking size, shape, and opening time, manifesting itself as a superior cloaking scheme.

Our paradigm may find practical applications in scenarios of robust digital communications[47,48] as well as secure communications[49,50]. In a practical digital communication system, the information to be transferred is usually carried by optical pulse trains. However, due to the presence of absorption and scattering events during propagation, the waveforms of pulse trains will get deformed, leading to the degradation of signal transmission quality. Here, by utilizing our temporal cloaking scheme, these unwanted events can be efficiently circumvented, enabling robust digital communications. Meanwhile, the absorption and scattering events can also act as spying signals through which the transmission signals can be monitored. The presence of temporal cloak can avoid such spying signals and hence could be adopted in secure communications. Finally, let's estimate the feasibility of our temporal cloak in a practical communication system. In our system, the pulse width and time interval between adjacent lattice sites are chosen as ~100 ns and ~75 ns, and the cloak width reaches ~45 lattice sites (Fig. 5d), corresponding to a broad time window of ~3375 ns. Such a cloak size can thus accommodate a typical temporal event in a low-speed

communication system at MHz ~ GHz ranges. By adjusting the modulation waveform, the size and shape of cloak can be further engineered, which further broads its application scenarios. Furthermore, to extend our temporal cloak to high-speed communication systems, the pulse width and lattice period should be scaled down to ps ~ ns ranges, accordingly. This could be realized by reducing the length difference between the two fiber loops and using high-speed optical and electronic devices in the circuit. Meanwhile, dispersion compensation techniques should be also utilized to reduce the influence of GVD for the short-pulse, high-speed systems.

## Discussion

In conclusion, we have shown that photonic quantum walks in synthetic temporal mesh lattices can provide a fertile platform to realize the Dunlap-Kenkre model of DL, and experimentally demonstrated different orders of DLs by applying artificial ac electric fields from sinusoidal phase modulations. By tuning the modulation amplitude to the roots of 0th-order Bessel function, we have realized the collapse of quasi-energy band structures and observed from first- up to fifth-order DLs, with a corresponding increase of localization strength over the full oscillation cycle. Remarkably, in the presence of external random noise, the higher-order DLs display much greater robustness against noise. The strategic quasi-energy band engineering enabled by our photonic platform can be harnessed to realize temporal cloaking. Specifically, we designed and experimentally demonstrated a temporal cloaking scheme by combining the higher- and lower-order DLs, with excellent tunability in terms of both cloaking window size and its opening time. It should be mentioned that the size of the temporal cloak is also not unlimited due to the presence of inevitable noise from optical amplifiers within the fiber loops. The tunability could be further optimized by using lower-power amplifiers with relatively lower noise, for which other lower-loss optical components should also be utilized. Because the generation of ac electric field is universal in fiber-optic systems, our demonstrated DL effect in temporal lattice can be transplanted to other synthetic dimensions, such as the frequency dimension of light[33–38].

Generally, the cloaking can be realized either by exploiting the wave packet's self-imaging effects in a uniform region or by introducing a time-reversal operation onto the wave packet in two cascading regions. For the former mechanism, there are two typical methods to achieve the packet's self-imaging, i.e., the effects of Bloch oscillations (BOs)[1,2,14,25] under a dc-field driving or our DL scheme under an ac-field driving. As discussed in details in Supplementary Note 9 and 10 and Supplementary Movie 1, our DL-based cloaking scheme has following advantages over BO-based one. Firstly, the BO-based cloak reproduces the shape of the band structure and hence can't be reshaped for a fixed lattice. By contrast, our DL-based scheme allows flexible reshaping of the cloak by varying the initial phase of ac driving even though the driving amplitude is fixed[51]. Furthermore, for BO- and DL-based cloaks with same oscillation period $M$, the DL's cloak is larger than that of BOs. For the time-reversal mechanism, the cloaking can be realized through negative refraction of two split beams. Compared with the DL scheme, the negative refraction requires negative refractive-index materials in real space[52,53], which is difficult to achieve. In synthetic dimensions, negative refraction is usually realized through an out-of-phase shift of gauge potentials in two cascading regions[34,54–56], which requires the precise and fixed utilization of modulation phases, and hence is not flexible and reconfigurable as the DL scheme. Finally, since our synthetic lattice is built fully in the temporal domain, the correponding cloaking is limited to one-dimensional systems and only applicable for temporal waveforms. However, if we combine the spatial cloaking methods with our temporal ones, a high-dimensional, composite spacetime cloak may also be envisaged[57,58].

As a final comment, it should be noted that the order of DLs is not unlimited due to onset of Landau-Zener tunneling[3,4,42] and even the breakdown of the lattice's continuous-time approximation under sufficiently large electric-field driving, corresponding to inevitable band mixing and the failure of continuous time-averaging picture for describing DLs. Specifically, for increasing values of the modulation amplitude $\Delta\phi$, one can qualitatively distinguish three different regimes (see Supplementary Note 2 for more detailed discussion): (i) For low-to-moderate modulation amplitudes, the continuous-time limit of the dynamics is valid and the field does not couple the two bands. In this regime we have DL like in the usual single-band Dunlap-Kenkre model, with quasi-energy band collapse observed at specific values of $\Delta\phi$, corresponding to the roots of $J_0$ Bessel function. (ii) For moderately high values of modulation amplitude, the ac field couples the two bands and nonadiabatic Landau-Zener tunneling is not negligible anymore. The quasi-energy band collapse is imperfect. (iii) At very high modulation amplitudes, such that $\Delta\phi\omega$ ceases to be smaller than 1, the discrete nature of temporal evolution cannot be disregarded anymore. In this regime, rather than the usual band flattening effect arising from the vanishment of Bessel function $J_0(\Delta\phi)$ for large $\Delta\phi$, one even observes an increase in quasi-energy bandwidth, resulting in dynamical delocalization. Hence, our discrete-time photonic quantum walk setup realizes a scenario of DL which can greatly deviate from the conventional Dunlap-Kenkre model of continuous-time systems, providing a fascinating platform to unravel a transition from continuous- to discrete-time localization features. For real applications, where noise and/or imperfections in the systems are unavoidable, the most stable operational regime includes the moderate- to high-order DLs, i.e., typically from 3rd to 6th orders. In this case, the localization is considerably tight and more robust against noise than low-order DL regime.

Finally, our results could inspire further theoretical and experimental studies as well as unravel new phenomena beyond the traditional Hermitian paradigm of discretized light transport. Thanks to the ability to engineer gain and loss in the synthetic mesh lattice setups[23,32], the correspondence between photonic quantum walks and the Dunlap-Kenkre model, unravelled in this work, could also pave the way toward the experimental demonstration of DL effects and related phenomena in the non-Hermitian realm. These include the experimental access to DL in parity-time systems[59] and the observation of non-Bloch band collapse and chiral tunneling in lattices displaying the non-Hermitian skin effect[30,60]. From the perspective of applications, due to the excellent tunability and strong robustness against noises, the high-order DLs we achieved could hold great promise for robust signal transmission, protection, storage, and processing.

## Methods

### Experimental setup and measurement

The light pulses are generated by modulating a continuous-wave light beam with an intensity modulator (IM), which is driven by the pulse from an arbitrary waveform generator (AWG). After being injected into the long loop, the pulse will circulate in the two loops. During circulation, the optical loss is compensated by erbium-doped fiber amplifiers (EDFAs). To overcome the transient of EDFA, the signal pulse is combined with a pilot light operated at a wavelength of 1530 nm. Then, the pilot light and spontaneous emission noise are removed by bandpass filters (BPFs). To detect the optical pulses, we couple the signals out of the loops and record them by photodiodes (PDs) and oscilloscopes (OSCs). The polarization states of light pulses are controlled through polarization controllers (PCs). A polarization beam splitter (PBS) and the subsequent PD are utilized to monitor the polarization state. Here, the phase modulation is only imposed in the short loop, which has a form of $-2\phi(m)$. The difference between the modulation phases in the two loops is $-2\phi(m)$, which is same with the one mentioned in the main text. Since not the phase but the phase difference

has a physical meaning, such a scheme of phase modulation is equivalent to the one mentioned in the main text. Finally, after ~200 circulations in the loops, all pulses are absorbed through switching off IMs. In addition, all the modulators are driven by AWGs. To realize the DL effect, a sinusoidally-varying phase modulation is introduced into the short loop. Since the DL arising from the collapse of the quasi-energy band structure happens only for a series of particular modulation amplitudes, one needs to carefully control the amplitude of the phase modulation.

### Preparation of Gaussian-envelope wave packet

To generate a Gaussian-envelope pulse sequence with specific Bloch momentum, we inject a single optical pulse into the long loop and impose phase and intensity modulations into the two loops[24,27]. By controlling the driving signals of IMs, the short loop is switched on and off alternately with the increase of the circulation number $m$ while the long loop stays on during the circulation. After 30 circulations, a pulse train with Gaussian envelope is formed. In addition, by imposing a constant phase shift $\alpha$ in the short loop during the 30 circulations, the Bloch momentum of wave packet can be chosen and has a form of $Q = (\pi - \alpha)/2$. To excite the eigen mode at the upper or lower band, we set phase and intensity modulations at the 31th circulation according to the eigen vector.

## Data availability

All the data supporting this study are available in the paper and Supplementary Information. Additional data related to this paper are available from the corresponding authors upon request.

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

## Acknowledgements

This work was supported by fundings from the National Natural Science Foundation of China (No. 11974124 (B.W.), 12021004 (P.L.), 12147151 (S.W.), and 12204185 (C.Q.)).

## Author contributions

B.W. conceived the idea. S.W., C.Q., and F.Z. designed and performed the experiment. S.W., C.Q., W.L., B.W., H.Y., and L.Z. analysed the data. C.Q., B.W., and S.L. provided theoretical support. B.W. and P.L. supervised the project. S.W., C.Q., W.L., B.W., P.L., S.L., F.Z., J.D., and X.Z.

contributed to the discussion of the results and writing of the manuscript.

## Competing interests

The authors declare no competing interests.
