## [Peer Review File · Nature Communications]

High-order dynamic localization and tunable temporal cloaking in ac-electric-field driven synthetic latticesREVIEWER COMMENTS

Reviewer #1 (Remarks to the Author):

I personally find the paper interesting, well written and suitable for publication after the following suggestions are considered and discussed.

I believe the paper is well presented and can provide useful to the audience of this journal but I fear the authors spent too little attention to several issues which I would like better discussed. I thus make the following suggestions:

1. Better analyze the effect of noise on the procedure. Consider more realistic noise models (other than the uniformly distributed noise already considered) and better quantify their effect.
2. Motivate the application to cloaking by better explaining its practical utility and importance.
3. provide a discussion about the issue of practical feasibility of the proposed application (for example cloaking method). Mention how practical is to instantiate and implement the proposed method discussing possible practical challenges and ways to overcome them.

Reviewer #2 (Remarks to the Author):

The article reports an experimental study of dynamical localisation (DL) of quantum particles whose dynamics take place in a one-dimensional lattice, dictated by nearest-neighbour couplings. Contrary to Anderson localisation, DL occurs in presence of an oscillating force. When the amplitude of the latter assumes one of a series of “magic values”, at times equal to integer multiples of the driving period the wavepacket goes back to its original position. Such magic values are labelled by an integer number.

As discussed by the authors, in the past DL has been observed in different settings, up to the second order. The appeal of accessing higher order DL is associated with an increasing localization of the wave packets, that in the first order case gets broader after multiple periods. However, this requires the ability to introduce strong drivings, remaining within adiabatic dynamics and preventing Landau-Zener transitions.

The experiment is performed by exploiting a standard optical system made of two coupled fiber-loops, that has been widely used to investigate the dynamics of non-interacting particles in 1D lattices (as acknowledged in the article literature). They simulate the effect of an oscillating force by tuning the value of a phase associated with couplings, whose signs depends on the particle hopping to the right or to the left in the temporal lattice. The authors show that high-order DL is actually affected by a lower spreading of the wavepacket, also when disorder is introduced to the system. Finally, by combining low- and high-order DL, the authors implement a temporal cloaking scheme, shaping light propagation in their synthetic lattice so that the wavepacket circumvent specific obstacles.

The paper is well written, the results and the theoretical background are convincing and presented clearly and extensively. Before providing a conclusive recommendation, I have some questions specifically targeted at understanding more deeply the potential impact of the reported research.

1) By looking at the supplemental material, it seems that there is no big increase in localization when moving from the second order to the fifth (see Fig. S2). In this sense, it can be questionable whether the ability to realize higher order DL could represent by itself a significant improvement of the system capability, besides being a novel and interesting result certainly deserving to be published. However, when going to the disordered configuration (see Fig. S4), there seems to be a much more significant difference between the first two orders and the higher order ones. Indeed, the latter actually yield a much more robust localisation. If the author confirm that this is the case, they could move the emphasis of their claims to this part.

2) Regarding the cloaking scheme, the ability to circumvent an obstacle seems to be associated with the breathing motion of particles, which can be observed also in the simpler case of Bloch oscillations. What are the key elements that make this approach significantly different? Is a setup based on DL achieving a cloaking action in cases where Bloch oscillations would fail?

Besides these major comments, below I list a series of minor remarks.

1) Can this approach be easily implemented in other platforms for photonic simulations, other than arrays of waveguides?

2) Is the process adiabatic? It would be beneficial for the readers having some comments on the constraints on the values of M , on the field amplitude and on the band gap that must be fulfilled to observe the desired DL.

3) In Fig. 3B, in panel (i) the wavepacket moves to the left, while in panel (ii) its average displacement is to the right. Intuitively, I would have expected an average movement to the left, as in panel (i). Perhaps a brief comment in the manuscript or in the figure caption that discuss the underlying mechanism could be beneficial.

4) In the concluding parts of the manuscript, it is discussed a possible effect of larger values of the field, that could lead to Landau-Zener tunnelings and, as a consequence, to a different dynamics. However, when increasing the field amplitude, can one expect a more pronounced band flattening? If yes, which is the role that Landau-Zener transitions would play? Wouldn't one observe any motion due to the absence of dispersion?

Reviewer #3 (Remarks to the Author):

The authors studied a coupled resonator system with unequal lengths, so a one-dimensional temporal mesh lattice is constructed by introducing train of pulses. The main goal of the manuscript is to study the dynamic localization effects in this mesh lattice. In addition, the authors claim that they demonstrate a temporal cloaking scheme. I am not convinced that this manuscript, at least in its current form, meet the high standards for publication in Nature Communications, for the reasons listed in the following:

1. The realization of DL in such mesh lattice is not surprising. Refer to the early work of observing the Bloch oscillation in the same mesh lattice [see Sci. Rep. 5, 17760]. Extending the realized effective DC electric field to AC electric field is not a big advance from the experimental point of view.
2. The dynamic localization has been explored in various platforms in experiments, and the current one does not show significant advance from the fundamental physical point of view. Of course, the high-order dynamic localization is novel. Nevertheless, such extension to realize the higher-order DL with the high-order zero solutions of Bessel function is well known, so I do not think that this novelty can guarantee the publication of the paper in NC.
3. The cloaking idea maybe attracting attentions at first glance. However, if one thinks it carefully, this so-called temporal cloaking is nothing but let the pulses avoid several sites in a one-dimensional mesh lattice when they are propagating versus the time and force them back near original positions later. Similar phenomena can be seen through different mechanisms in optics, such as a time-reversal operation (which by the way, I believe that the authors can do this operation too in the current setup without the use of DL). The studied temporal cloaking has no benefit to developments of the optical cloaking in real space.

Response Letter to Manuscript ID: NCOMMS-22-10301

High-order dynamic localization and tunable temporal cloaking in ac-electric-field driven synthetic lattices

by Shulin Wang, Chengzhi Qin, Weiwei Liu, Bing Wang, Feng Zhou, Han Ye, Lange Zhao, Jianji Dong, Xinliang Zhang, Stefano Longhi and Peixiang Lu

Reviewer 1:

Comments:

I personally find the paper interesting, well written and suitable for publication after the following suggestions are considered and discussed. I believe the paper is well presented and can provide useful to the audience of this journal but I fear the authors spent too little attention to several issues which I would like better discussed. I thus make the following suggestions.

Reply: We thank the referee for his/her positive evaluation on our work.

[Comment 1]

Better analyze the effect of noise on the procedure. Consider more realistic noise models (other than the uniformly distributed noise already considered) and better quantify their effect.

Reply: We thank the reviewer for this relevant comment. Previously, we focused on the influence of a **uniformly distributed noise** on high-order dynamic localizations (DLs) and found that high-order DLs have much better robustness compared to low-order ones. Here we choose two other mostly-studied noise models **with Gaussian and gamma distributions** to further demonstrate the enhanced robustness of high-order DLs. The simulation results show that the enhanced robustness of high-order DLs is universal for different types of noise models.

Firstly, for a modulation-phase noise ϕ_{noise} with Gaussian distribution, its probability density function is given by

$$P(\phi_{\text{noise}}) = \frac{1}{\sqrt{2\pi\sigma^2}} e^{-(\phi_{\text{noise}} - \mu)^2 / 2\sigma^2}, \quad (\text{R1})$$

where μ and σ are the expected value and standard deviation of the random noise, respectively. As plotted in the inset of Fig. R1a, the probability density function possesses a Gaussian shape. Here we choose $\mu = 0$ and $\sigma = 0.043\pi$, both of which equal to those of the uniformly distributed noise used in the main text. To study the influence of Gaussian noise on DLs, we simulate the wave packet dynamics under 5000 realizations of Gaussian noises and calculate the averaged inverse participation ratio (*IPR*) at output step $m = 120$, as depicted in Fig. R1a. One sees that the *IPR* versus the modulation amplitude $\Delta\phi$ is almost identical to that of a uniform noise (see Fig. 4b in the main text). The value of *IPR* peaks at each order of DLs, indicating where the noise-induced wave packet spreading should be suppressed. Besides, the *IPR* increases as the DL order increases, reflecting that the high-order DL manifests better

robustness against the Gaussian noise. Figures R1b(i)-R1b(iv) illustrate the simulated pulse intensity evolutions for $\Delta\phi = 0, 2.4$ (1st-order DL), 8.7 (3rd-order DL) and 14.9 (5th-order DL), respectively. For the case without external field ($\Delta\phi = 0$), the wave packet spreads in a wide range of direction under the noise. As the DL occurs, the spreading of the wave packet is highly suppressed by comparing with the field-free case. As the order of DL increases, the suppression becomes more remarkable. Especially for the 5th-order DL, the width of wave packet almost keeps constant during evolution.

Fig. R1. **a** Inverse participation ratio IPR of wave packet at $m = 120$ in the presence of Gaussian noise. The inset displays the probability density function of Gaussian noise. **b** Simulated pulse intensity evolutions for (i) $\Delta\phi = 0$. (ii)-(iv) The 1st, 3rd- and 5th-order DLs, respectively. **c** Inverse participation ratio IPR of wave packet at $m = 120$ under gamma noise. The probability density function of gamma noise is shown in the inset. **d** Simulations of averaged pulse intensity evolutions.

Next we consider the gamma noise with the probability density function

$$P(\phi_{noise}) = \frac{1}{b^a \Gamma(a)} \phi_{noise}^{a-1} e^{-\phi_{noise}/b} - ab, \quad (R2)$$

where a and b are two positive parameters determining the shape of gamma distribution, and $\Gamma(a)$ is the gamma function. Note that the expected value of the noise is zero, and the standard deviation equals $a^{1/2}b$. Here, we choose $a = 1.85$ and $b = 0.1$ to ensure the standard deviation of noise remains the same level with the cases of uniform and Gaussian noise. As depicted in the inset of Fig. R1c, the random phase manifests a gamma distribution. In the presence of gamma noise, we calculate the averaged inverse participation ratio (IPR) at $m = 120$ in Fig. R1c. One sees clearly that the enhanced robustness of high-order DLs is also well preserved for gamma noise, and the dependence of IPR on $\Delta\phi$ is identical to these of the uniform and Gaussian cases. In Figs. R1d(i)-R1d(iv), the simulated pulse intensity evolutions also verify the above analysis. Hence, the enhanced robustness of high-order DL is universal for more realistic noise models.

In the revised main text, we added the discussion about the influence of different-type noises on the DL dynamics and clarified that the superior robustness of high-order DLs is a universal behavior for different-type noises. In experiment, we choose the uniformly distributed noise as a representative example. Please refer to the paragraph before Eq. (9) and the sentence 2, paragraph 2, page 10 in the main text and the inserted figure in Fig. 4b showing the probability density function of uniform noise. The simulation results for the two other representative noises of Gaussian and gamma distributions are added in Supplementary Note 7.

[Comment 2]

Motivate the application to cloaking by better explaining its practical utility and importance.

Reply: Thanks for this suggestion. Let's firstly outline some practical scenarios for temporal cloaking and then clarify the importance of our cloaking scheme by comparing with previously adopted schemes.

(1) One of the most important applications of the temporal cloaking is to improve the signal transmission quality in digital communication systems^{1,2}. In practical digital systems, the information to be transferred is often carried by a pulse train consisting of certain optical pulses. However, the disturbances during the transmission process, including absorption and scattering, may damage the waveform of the pulse train and hence degrade the quality of the signal transmission. By utilizing the temporal cloaking, the absorption or scattering event can be perfectly circumvented, which preserve

the pulse train during propagation and thus significantly improve the quality of signal transmission.

(2) For another application purpose, the temporal event can also be treated as a spying signal, and one can spy the transmitting information through inspecting the scattered or absorbed optical signal caused by the temporal event. The opening of the temporal cloak avoids the interaction between the transmitting pulse train and the spying signal^{3,4}. Hence, the information carried by the pulse sequence can be concealed by using such a scheme, which has great applications in secure communications.

(3) Our temporal cloaking scheme shows significant advantages over the traditional dispersion-based temporal cloaking mechanism. Previously, the temporal cloaking is achieved in optical fibers where the normal and anomalous group velocity dispersions (GVDs) of optical pulses contribute to the opening and closing of temporal cloak^{5,6}. Limited by the fixed GVD coefficient of optical fiber, the cloaking size and opening time are difficult to manipulate, which highly constrains the practical application of temporal cloak. Here, the temporal cloak is realized through combination of lower- and higher-order DLs using strategic phase modulations. By tuning the phase modulation in full dynamic manners, one can flexibly manipulate the cloaking size and opening time, which greatly enhances the capabilities and reconfigurabilities for practical application scenarios.

In the revised manuscript, we added the discussion about the practical utility and its advantage in paragraph 1, page 13 and lines 1-9, paragraph 2, page 13. The relevant references have also been added into the main text, see Refs. 45-50.

Related references in the above discussions:

1. Lukens, J. M., Metcalf, A. J., Leaird, D. E. & Weiner, A. M. Temporal cloaking for data suppression and retrieval. *Optica* **1**, 372-375 (2014).
2. Cortés, L. R., Seghilani, M., Maram, R. & Azaña, J. Full-field broadband invisibility through reversible wave frequency-spectrum control. *Optica* **5**, 779-786 (2018).
3. Bony, P. Y. *et al.* Temporal spying and concealing process in fibre-optic data transmission systems through polarization bypass. *Nat. Commun.* **5**, 1-9 (2014).
4. Zhou, M., Liu, H., Sun, Q., Huang, N. & Wang, Z. Temporal cloak based on tunable optical delay and advance. *Opt. Express* **23**, 6543-6553 (2015).
5. Fridman, M., Farsi, A., Okawachi, Y. & Gaeta, A. L. Demonstration of temporal cloaking. *Nature* **481**, 62-65 (2012).
6. Lukens, J. M., Leaird, D. E. & Weiner, A. M. A temporal cloak at telecommunication data rate. *Nature* **498**, 205-208 (2013).

[Comment 3]

Provide a discussion about the issue of practical feasibility of the proposed application (for example cloaking method). Mention how practical is to instantiate and implement the proposed method discussing possible practical challenges and ways to overcome them.

Reply: (1) **The application feasibility of our cloaking scheme.** In our experiment, the pulse width is chosen as ~ 100 ns, and the time interval between adjacent lattice sites is ~ 75 ns. For the driving period $M = 120$ and the coupling ratio 75:25, the cloak width approaches 45 lattice sites, which corresponds to a ~ 3375 ns broad time window. The cloak size can thus accommodate a typical temporal event in a low-speed communication system at MHz \sim GHz ranges. By programming the arbitrary waveform generator that drives the phase modulator, we can configure the period of phase modulation and hence adjust the cloak width to adapt different application scenarios in practice. In experiment, we have realized a series of temporal cloaks with their widths ranging from ~ 1125 ns to ~ 3375 ns. More importantly, such a cloak can be dynamically modified by tuning the output signal of the arbitrary waveform generator, which greatly improves the feasibility of the cloaking scheme.

(2) **The possible practical challenges and ways to overcome them.** To extend our cloaking scheme to the current high-capacity digital communication systems, much shorter pulses of ns \sim ps scales need to be utilized to carry numerous digital bits. Firstly, the short pulse's broadening caused by GVD cannot be ignored since the broadening can reach several nanoseconds. To deal with this issue, one can insert dispersion compensated fibers into the system. Secondly, the short pulse possesses much higher peak power compared with the long pulse currently in use, which may induce optical nonlinear effects and even damage the optical devices in the system. Under the circumstances, the incident pulse train should be appropriately attenuated before entering the system. Finally, high-speed optical and electronic devices should also be utilized to meet the bandwidth requirement during the modulation and detection of optical pulses.

For the revision in the main text, we added the discussion about the feasibility of our temporal cloak in lines 9-19, paragraph 2, page 13.

Reviewer 2:

Comments:

The article reports an experimental study of dynamical localisation (DL) of quantum particles whose

dynamics take place in a one-dimensional lattice, dictated by nearest-neighbour couplings. Contrary to Anderson localisation, DL occurs in presence of an oscillating force. When the amplitude of the latter assumes one of a series of “magic values”, at times equal to integer multiples of the driving period the wavepacket goes back to its original position. Such magic values are labelled by an integer number.

As discussed by the authors, in the past DL has been observed in different settings, up to the second order. The appeal of accessing higher order DL is associated with an increasing localization of the wave packets, that in the first order case gets broader after multiple periods. However, this requires the ability to introduce strong drivings, remaining within adiabatic dynamics and preventing Landau-Zener transitions.

The experiment is performed by exploiting a standard optical system made of two coupled fiber-loops, that has been widely used to investigate the dynamics of non-interacting particles in 1D lattices (as acknowledged in the article literature). They simulate the effect of an oscillating force by tuning the value of a phase associated with couplings, whose signs depends on the particle hopping to the right or to the left in the temporal lattice. The authors show that high-order DL is actually affected by a lower spreading of the wavepacket, also when disorder is introduced to the system. Finally, by combining low- and high-order DL, the authors implement a temporal cloaking scheme, shaping light propagation in their synthetic lattice so that the wavepacket circumvent specific obstacles.

The paper is well written, the results and the theoretical background are convincing and presented clearly and extensively. Before providing a conclusive recommendation, I have some questions specifically targeted at understanding more deeply the potential impact of the reported research.

Reply: We thank the referee for his/her positive evaluation on our work.

[Comment 1]

By looking at the supplemental material, it seems that there is no big increase in localization when moving from the second order to the fifth (see Fig. S2). In this sense, it can be questionable whether the ability to realize higher order DL could represent by itself a significant improvement of the system capability, besides being a novel and interesting result certainly deserving to be published. However, when going to the disordered configuration (see Fig. S4), there seems to be a much more significant difference between the first two orders and the higher order ones. Indeed, the latter actually yield a

much more robust localisation. If the author confirm that this is the case, they could move the emphasis of their claims to this part.

Reply: (1) We thank the reviewer for this interesting comment. At first glance, the localization strength increase is not sufficiently obvious from the 2nd to 5th orders. But when checking the mean-square displacement $\langle n^2(m) \rangle_{\max}$ quantitatively, we find the localization strength shows appreciable increase as the order increases. When moving from the 2nd to the 4th and 5th orders, the packet maximum mean-square displacement $\langle n^2(m) \rangle_{\max}$ reduces by 31% and 21%, respectively, which already indicates a considerable enhancement of localization strength, even not obvious as the reduction of 61% from the 1st- to 2nd-order DLs.

(2) Furthermore, the improvement of localization strength is directly proportional to the ac driving period M . In our experiment, we choose a relatively small driving period of $M = 120$, such that the localization strength enhancement is not so obvious from 2nd to 5th orders. Here we choose a relatively larger driving period $M = 400$ and numerically verify that the decrease of $\langle n^2(m) \rangle_{\max}$ can be improved. As shown in Fig. R2a, $\langle n^2(m) \rangle_{\max}$ decreases from 124.3 to 65.4 from the 2nd- to 4th-order DLs, indicating a significant reduction of 47%. As also shown in Figs. R2b-d, the 4th- and 5th-order DLs exhibit more localized oscillation ranges compared to 2nd-order DL, further verifying the localization strength enhancement at a long driving period. In our current setup of fiber-loop experiment, due to the accumulated noise from optical amplifiers, it is not easy to observe 400 or even more circulation times for the optical pulse at a sufficiently low noise level. Under the circumstances, we only show the experimental results for $M = 120$ in this work. We added the relevant discussion into the revised version, please refer to sentences 5 and 6 of the paragraph before the section “Enhanced robustness of high-order DLs against random noises” in the main text and Supplementary Note 5.

(3) When introducing stochastic noise into the artificial electric field, the high-order DLs exhibit much better robustness against the random noise compared to the low-order ones. More importantly, the enhancement of the robustness from the low- to high-order DLs is much more obvious compared to the increase of the localization strength. Thus, as properly suggested by the Referee in the revised manuscript we emphasized the enhanced robustness of the high-order DL. Please refer to the last two sentences, paragraph 1 of the introduction part to clarify our claim and the revised paragraph 1 of section “Enhanced robustness of high-order DLs against random noises” in the main text.

Fig. R2. **a** Maximum mean-square displacement $\langle n^2(m) \rangle_{\max}$ versus order of DL for driving period $M = 400$. **b-d** Simulated pulse intensity evolutions for 2nd-, 4th- and 5th-order DLs.

[Comment 2]

Regarding the cloaking scheme, the ability to circumvent an obstacle seems to be associated with the breathing motion of particles, which can be observed also in the simpler case of Bloch oscillations. What are the key elements that make this approach significantly different? Is a setup based on DL achieving a cloaking action in cases where Bloch oscillations would fail?

Reply: (1) We thank the Referee for suggesting this very interesting point. As correctly stated by the Referee, in principle the cloaking scheme could be implemented and observed exploiting the breathing motion associated to Bloch oscillations (BOs), i.e. using a dc electric field rather than an ac field. However, the ac field case offers a much more flexible design to shape the cloaking region. In our experiments we already showed that the cloak possesses flexible reconfigurability both in the cloaking window size and its opening time. Even more exciting, for a given fixed driving period the cloaking region can be dynamically reshaped. In fact, the shape of the cloaking region is basically given by the area enclosed by the paths followed by the two split beams, which can be estimated from the group velocities corresponding to two different bands

$$v_{g,\pm}(m) = \mp \cos(\beta) \sin[Q(m)] = \mp \cos(\beta) \sin[Q - \phi(m)]. \quad (\text{R3})$$

For a dc field α , leading to BOs, the momentum $Q(m) = Q + am$ changes linearly in time, so that the path followed by the beam exactly reproduces the shape of band structure. The main limits of using the BO approach (or other similar approaches such as time reversal) is that we cannot reshape the

cloaking region: its shape always reproduces the shape of the energy dispersion curve. On the other hand, using an ac field we can control and dynamically reconfigure the path of the beam, for a fixed value of the driving period. By contrast, it is known that the path followed the quantum particle in the crystal under an ac field does not reproduce the shape of the dispersion curve and can be dynamically controlled, for example, by the initial phase of the ac field¹. In particular, for a sinusoidal field $E_{eff}(m) = \omega\Delta\phi\sin(\omega m + \varphi)$, as in our experiment, by varying the phase φ of the field (from 0 to π) we can largely deform the beam path, i.e. reshaping the cloaking region. This is shown as an example in the attached movie (submitted as an attachment together with the revised manuscript), where we compare the beam paths under BOs (for a dc field $\alpha = \omega$) and under DL for increasing values of the phase φ . Parameter values are $\beta = \pi/3$, $\Delta\phi = 2.4$, $\omega = \pi/60$. Clearly, the movie shows that, at a fixed value of the duration time (defined by the period $M = 2\pi/\omega$), the beam path in the DL regime can be largely tailored by changing the phase of the sinusoidal field.

For a non-sinusoidal ac field, an even more flexible cloaking region reshaping could be realized. As an illustrative example, in Fig. R3 we show how the paths of wave packets (and thus the cloaking region) can be shaped by changing the ac field $\Delta\phi(m)$. The only requirement to observe self-imaging after one oscillation cycle is provided by the DL condition

$$\Delta n_{\pm} = \int_0^M v_{g,\pm}(m) dm = \mp \int_0^M \cos(\beta) \sin[Q - \phi(m)] dm = 0 \quad (\text{R4})$$

which reduces to the usual condition of $J_0(\Delta\phi)=0$ for the sinusoidal field, as demonstrated by our work. Panel (a) reproduces our experiment, where we used a sinusoidal field at first-order DL point, i.e. $\Delta\phi(m) = \Delta\phi\cos(\omega m + \varphi)$ with $\Delta\phi = 2.4$, $\varphi = \pi/2$, $\omega = \pi/60$, $\beta = \pi/3$. Panel (b) corresponds to a modified ac field shaped as $\Delta\phi(m) = \Delta\phi_1\cos(\omega m + \varphi) + \Delta\phi_2\sin^3(\omega m + \varphi)$ with $\Delta\phi_1 = 1$, $\Delta\phi_2 = 3.1$, $\varphi = 2.5$, $\omega = \pi/60$, $\beta = \pi/3$.

These are just two illustrative examples; however they clearly demonstrate how the cloaking region can be reconfigured and dynamically shaped by simply acting on the signal voltage driving the phase modulators in the fiber loops. Our experiment thus proves the supremacy of the cloaking scheme based on the DL regime over other schemes, such as those based on BOs as suggested by the Referee. Dynamical reshaping of the cloaking region enabled by the high/low reconfigurable DL switching scheme is a powerful functionality which could be of major relevance in cloaking applications. Therefore, we fully believe that future developments of the reconfigurable optical cloaking in real fiber-based devices could greatly benefit from the scheme suggested and demonstrated in our

experiment.

Fig. R3. Example of reconfigurability of the cloaking region in the mesh lattice setup by change of the ac field shape. In **a** $\Delta\phi(m) = \Delta\phi\cos(\omega m + \varphi)$, in **b** $\Delta\phi(m) = \Delta\phi_1\cos(\omega m + \varphi) + \Delta\phi_2\sin^3(\omega m + \varphi)$.

(2) Different from the BO-based cloak, the DL-based cloak also holds the advantage in the cloak width. Under a dc field α , the transverse group velocities for lower and upper bands read

$$v_{g,\pm}(m) = \mp \cos(\beta) \sin(Q + \alpha m - \phi_2), \quad (\text{R5})$$

As $Q = \pi/2$ and $\phi_2 = \pi/2$, the group velocities reduce to

$$v_{g,\pm}(m) = \mp \cos(\beta) \sin(\alpha m). \quad (\text{R6})$$

At the first half of the Bloch period $M = 2\pi/\alpha$, the group velocities corresponding to the lower and upper bands are positive and negative, respectively. As a consequence, the wave packets belonging to the lower and upper bands continuously move to the rightward and leftward, respectively, leading to the opening of a time window. At the second half of Bloch period, the wave packets' group velocities become opposite to the first ones, and the time window is gradually closed. The maximum width of the time window, i.e. the cloak width, can be calculated as

$$W_{\max} = \int_0^{M/2} [v_{g,-}(m) - v_{g,+}(m)] dm = \frac{2 \cos(\beta)}{\pi} \cdot M. \quad (\text{R7})$$

Figure R4a depicts the cloak width W_{\max} with respect to the period M for BOs and DL. One sees that the widths of BO- and DL-based cloaks are both directly proportional to the oscillation period M of BOs and DL. Furthermore, for an identical oscillation period M , the DL-based cloak has a broader width, promising a better capability in practical applications. The differences in the cloak width can

be explained by the group velocities of wave packets displayed in Figs. R4b and R4c. At the first half of the oscillating period, the velocity difference between two wave packets that belong to lower and upper band makes the two wave packets separated and results in the maximum separation at $m = M/2$. The cloak width could be calculated by adding the areas of blue and red regions that denote the displacements of two wave packets. For DL, the colored region is obviously larger than the one of BOs, which reflects a bigger cloak width. Figures R4d and R4e display the pulse intensity evolutions of DL- and BO-based temporal cloaks with a period of $M = 120$. The DL-based cloak has a broader width $W_{\max} = 45$ while the BO-based one corresponds a smaller cloak width of $W_{\max} = 38.2$.

Thanks for the reviewer's suggestions, the above discussion was added in the second paragraph of the discussion part in the main text, Supplementary Note 9 and 10, Ref. 51 and attached movie.

Related reference in the above discussion:

1. Longhi, S. *et al.* Semiclassical motion of a multiband Bloch particle in a time-dependent field: Optical visualization. *Phys. Rev. B* **74**, 155116 (2006).

Fig. R4. **a** Cloak width W_{\max} varying with period M for DL and BOs. **b, c** Transverse group velocity v_g versus step m for DL and BOs. The area of blue (red) region represents the displacement of wave packet for the lower (upper) band after the first half of the period M . **d, e** Simulated pulse intensity evolutions for DL- and BO-based temporal cloaks.

[Comment 3]

Can this approach be easily implemented in other platforms for photonic simulations, other than arrays of waveguides?

Reply: The vector-potential-based artificial ac electric field and high-order DLs can also be realized by utilizing the **synthetic frequency dimension** constructed from dynamical modulation. For example, a dynamic refractive-index modulation in a straight waveguide induces the transition between a series of uniformly-distributed frequency modes as phase-matching condition is satisfied^{1,2}. These coupled frequency modes mimic a lattice model born in synthetic frequency dimension, where the waveguide's propagation direction plays the role of the evolution time in traditional spatial lattice models. In addition, the initial modulation phase could lead to a direction-dependent phase during the frequency transition and produce an artificial vector potential in the frequency lattice. By introducing a slight wave vector mismatch during transitions, one can further obtain a time-varying vector potential and hence an ac electric field. Like in our DL's scheme, the DL effect in the frequency can also be achieved by choosing appropriate amplitude of ac-driven electric field. In addition, because the synthetic frequency lattice does not contain any sublattice in a single lattice site, there only exists one band in the band structure. The Landau-Zener tunneling effect can be avoided automatically, and hence even higher-order DLs are capable to be observed. Apart from the single straight waveguide, the synthetic frequency lattice can also be generated from a dynamically-modulated fiber ring, where the modulation frequency equal to the ring's free spectral range. Such a phase modulation also induces the coupling between a series of resonant modes and leads to a synthetic frequency lattice. By introducing a time-varying detuning in the modulation frequency, the time-varying vector potential and ac electric field also emerge in this lattice and result in the frequency DL effect³⁻⁶. The relevant papers have been added into the references, see Refs. 33-38. See also the last sentence, first paragraph of discussion part for the added discussion.

Related references in the above discussion:

1. Qin, C., Yuan, L., Wang, B., Fan, S. & Lu, P. Effective electric-field force for a photon in a synthetic frequency lattice created in a waveguide modulator. *Phys. Rev. A* **97**, 063838 (2018).
2. Qin, C. *et al.* Spectrum control through discrete frequency diffraction in the presence of photonic gauge potentials. *Phys. Rev. Lett.* **120**, 133901 (2018).

3. Yuan, L. & Fan, S. Bloch oscillation and unidirectional translation of frequency in a dynamically modulated ring resonator. *Optica* **3**, 1014-1018 (2016).
4. Dutt, A. *et al.* Experimental band structure spectroscopy along a synthetic dimension. *Nat. Commun.* **10**, 3122 (2019).
5. Wang, K., Dutt, A., Wojcik, C. C. & Fan, S. Topological complex-energy braiding of non-hermitian bands. *Nature* **598**, 59-64 (2021).
6. Wang, K. *et al.* Generating arbitrary topological windings of a non-hermitian band. *Science* **371**, 1240-1245 (2021).

[Comment 4]

Is the process adiabatic? It would be beneficial for the readers having some comments on the constraints on the values of M , on the field amplitude and on the band gap that must be fulfilled to observe the desired DL.

Reply: We thank the Referee for this interesting comment and suggestion. To realize the conditions of DL, in our experiments we assumed that the time evolution is adiabatic, so that the time step variable m can be basically considered as a continuous variable. Adiabaticity clearly implies some constraint on the modulation frequency ω and amplitude $\Delta\phi$ of the field. In this regime quasi-energy band collapse basically gives the same condition of DL like in the original Dunlap-Kenkre model of DL (zeros of Bessel function), as we discussed in Supplementary Note 2. The adiabaticity of evolution is satisfied provide that $\Delta\phi(m)$ changes by a small quantity as m changes by ± 1 . For a sinusoidal driving $\phi(m) = \Delta\phi\cos(\omega m + \varphi)$, this condition is satisfied provided that (i) the oscillation period $M = 2\pi/\omega$ is large enough and, (ii) the amplitude $\Delta\phi$ is smaller than $1/\omega$. A more detailed study on the validity and breakdown of the adiabaticity condition can be gained by numerically computing the exact quasi-energy spectrum of the quantum walk system, beyond the averaging method given by Eq. (20) in the supplemental document. In the revised manuscript, we added a new subsection to Supplementary Note 2, where we provide exact numerical results of quasi-energy spectrum and discuss the breakdown of adiabaticity. Specifically, we show that for high field amplitudes quasi-energy band collapse is not anymore observed at high-order zeros of Bessel function, as a result of the breakdown of adiabaticity, the onset of band coupling (Zener tunneling) and discreteness nature of time evolution. Also, contrary to the typical scenario found in ac-driven continuous-time systems, quasi-energy band flattening is not

any more observed at large modulation amplitudes $\Delta\phi$. For a more extended discussion on this point, see also our reply to comment 6.

[Comment 5]

In Fig. 3B, in panel (i) the wavepacket moves to the left, while in panel (ii) its average displacement is to the right. Intuitively, I would have expected an average movement to the left, as in panel (i). Perhaps a brief comment in the manuscript or in the figure caption that discuss the underlying mechanism could be beneficial.

Reply: Thanks for the reviewer's suggestion. The wave packet's average movement to the left usually corresponds to a negative averaged group velocity and hence a negative displacement. As displayed in Fig. 3a, the leftward motion can be realized by choosing the modulation amplitude $\Delta\phi$ within several specific ranges, such as $[0, 2.4)$ and $(5.5, 8.7)$. In order to inspect the instantaneous dynamics of wave packet, we depict the instantaneous group velocity of wave packet $v_g(m)$ for $\Delta\phi = 1.7$ in Fig. R5a where the areas of green and blue regions represent the leftward and rightward shifts, respectively. It shows that the leftward shift is much larger than the rightward one, reflecting a net leftward shift after the driving period. In addition, within the driving period, the alternate change of motion direction indicates oscillations of wave packet during propagation. Figure R5b shows the corresponding simulated pulse intensity evolution. One can see that the wave packet moves to the left accompanying with a slight oscillating trajectory, which coincides well with the above analysis. In the revised manuscript, we added a brief discussion on the underlying mechanism. Please refer to the last sentence in the first paragraph of page 8.

Fig. R5. a Instantaneous group velocity v_g versus step m for $\Delta\phi = 1.7$. The area of green (blue) region represents the negative (positive) displacement of wave packet. **b** Simulated pulse intensity evolution.

[Comment 6]

In the concluding parts of the manuscript, it is discussed a possible effect of larger values of the field, that could lead to Landau-Zener tunnelings and, as a consequence, to a different dynamics. However, when increasing the field amplitude, can one expect a more pronounced band flattening? If yes, which is the role that Landau-Zener transitions would play? Wouldn't one observe any motion due to the absence of dispersion?

Reply: The picture suggested by the Referee, i.e. quasi-energy band flattening at high values of the ac field amplitude, is the typical scenario that one observes in the ordinary DL problem of a quantum particle hopping on a nearest-neighbor tight-binding lattice with a superimposed sinusoidal field **in the continuous-time limit**, i.e. in the usual Dunlap-Kenkre model of DL in a single band lattice. In the two-band case, we have of course to include Zener tunneling for high fields (see e.g. Ref. 1 below for a detailed analysis of Zener tunneling in continuous-time ac-driven two-band systems), however as correctly stated by the Referee at very high field amplitudes we have quasi-energy band flattening and basically the dynamics is frozen.

Fig. R6. a Quasi-energy spectrum $\varepsilon(Q)$ versus modulation amplitude $\Delta\phi$ calculated by the average

method under the continuous-time limit. **b** Exact quasi-energy spectrum obtained from numerical computation. The parameter values are $\omega = \pi/30$, $\beta = 0.97 \times \pi/2$. Only the upper quasi-energy band is plotted. The right panel shows an enlargement of the quasi-energies near the 10th root of J_0 Bessel function (vertical dashed line).

Remarkably, the scenario for high field amplitudes is very different when dealing with a **discrete-time** photonic quantum walk system: rather surprising, the frozen dynamics picture suggested by the Referee at high modulation amplitudes is not observed anymore and one typically observes dynamical delocalization, i.e. the absence of localization despite the very high amplitude of the ac field. Why do we miss to observe the frozen dynamics and band flattening phenomenon in the discrete-time photonic quantum walk setup? The photonic quantum walk on a mesh lattice basically reduces to the continuous-time Dunlap-Kenkre model only in the adiabatic limit of slow modulation, where the phase $\phi(m) = \Delta\phi \cos(\omega m + \varphi)$ varies slowly with the discrete-time index m . In our experiment we assumed in fact a low oscillation frequency ($\omega = \pi/60$). For increasing values of the modulation amplitude $\Delta\phi$, one can qualitatively distinguish three different regimes: (i) For low-to-moderate modulation amplitudes, the continuous-time limit of the dynamics is valid and the ac field does not couple the two bands. In this regime we have DL like in the usual single-band Dunlap-Kenkre model, with quasi-energy band collapse observed at specific values of $\Delta\phi$, corresponding to the roots of the J_0 Bessel function. (ii) For moderately high values of the modulation amplitude, the coupling of two bands occurs and Zener tunneling is not negligible anymore. The quasi-energy band collapse is imperfect. (iii) At very high modulation amplitudes $\Delta\phi$, such that $\Delta\phi\omega$ ceases to be smaller than 1, even for a slow oscillation frequency the phase $\phi(m)$ does not change slowly with index m , and thus the discrete nature of temporal evolution cannot be disregarded anymore and we cannot use the usual band flattening result known for continuous-time two-band ac-driven systems. As a matter of fact, in the ultrastrong amplitude regime we need to numerically compute the exact quasi-energy bands of the discrete-time mesh lattice. Figure R6 shows, as an illustrative example, the behavior of quasi-energies (only the upper branch is plotted) of the discrete-time mesh-lattice versus modulation amplitude $\Delta\phi$ for parameter values $\omega = \pi/30$, $\beta = 0.97 \times \pi/2$, as obtained from the exact numerical computation of Floquet exponents of the one-period propagator of the system (Fig. R6b), and from the relation $\varepsilon(Q) = -J_0(\Delta\phi)\cos(\beta)\cos(Q) + \pi/2$

corresponding to the averaging method discussed in the main manuscript (Fig. R6a). Note that, for low to intermediate values of the modulation amplitude $\Delta\phi$, quasi-energy band collapse at the various DL orders (zeros of J_0 function) is observed, in agreement with the Dunlap-Kenkre model. However, at high values of the modulation amplitude $\Delta\phi$, quasi-energy band collapse is not exact (see for example the case $\Delta\phi = 30.6$, corresponding to the 10th root of Bessel function, shown in the right panels of Fig. R6). Remarkably, unlike the Dunlap-Kenkre model the width of the quasi-energy band does not shrink to zero at very high values of the field amplitude $\Delta\phi$, and displays an irregular behavior (Fig. R6b). This means that, unlike continuous-time ac driven systems, the dynamics is not frozen and localization is not observed in the ultrastrong field amplitude regime. This is shown, as an example, in Fig. R7. The figure depicts the numerically-computed spreading of an initial single-site excitation in the mesh lattice system in the ultrastrong field regime, for a coupling angle and modulation frequency which are the same as used in our experiment ($\omega = \pi/60$, $\beta = \pi/3$, $\Delta\phi = 91$). Note that, in spite of the very strong amplitude $\Delta\phi$ of the field, DL is not observed. This is a strict feature of the DL in the photonic quantum walk system on a mesh lattice, arising from the discrete nature of time evolution.

In the revised manuscript, we clarified such points in the last two sentences before Eq. (4) and also paragraph 3 of the discussion part. In the supplementary notes, we also added a new section and a new figure in section C of Supplementary Note 2.

Fig. R7. Delocalization in the mesh lattice system under a ultrahigh-field ac sinusoidal driving ($\omega = \pi/60$, $\beta = \pi/3$, $\Delta\phi = 91$). The lattice is excited at the initial time step $m = 0$ in the single site $n = 0$.

Related reference in the above discussion:

1. Longhi, S. Bloch-Zener quantum walk. *J. Phys. B: At., Mol. and Opt. Phys.* **45**, 225504 (2012).

Reviewer 3:

Comments:

The authors studied a coupled resonator system with unequal lengths, so a one-dimensional temporal mesh lattice is constructed by introducing train of pulses. The main goal of the manuscript is to study the dynamic localization effects in this mesh lattice. In addition, the authors claim that they demonstrate a temporal cloaking scheme. I am not convinced that this manuscript, at least in its current form, meet the high standards for publication in Nature Communications.

Reply: We thank the Referee for reviewing our manuscript and for frankly expressing his/her view about the suitability/unsuitability for publication of our work in Nature Communications. We respectfully disagree with the main Referee's criticisms, and we think that the Referee's opinion was very likely motivated by the lack of clarity in explaining the novelty and major advances of our experimental results in the previous version of the manuscript. We have now greatly revised our manuscript to overcome and/or rebut all Referee's criticisms.

[Comment 1]

The realization of DL in such mesh lattice is not surprising. Refer to the early work of observing the Bloch oscillation in the same mesh lattice [see Sci. Rep. 5, 17760]. Extending the realized effective DC electric field to AC electric field is not a big advance from the experimental point of view.

Reply: (1) We agree and properly acknowledge in our paper that the mesh lattice setup has been previously used for observing Bloch oscillations (BOs). We now properly acknowledged such a previous work and added the reference Sci. Rep. 5, 17760 in the main text, see Ref. 25. However, dynamic localization (DL) is a different physical phenomenon than BOs and, to the best our knowledge, this phenomenon has been not yet demonstrated in any discrete-time photonic quantum walk setup. We reveal that the mesh lattice system can, in the slow (adiabatic) regime, exactly reproduce the original Dunlap-Kenkre model, enabling for the observation on high-order DL. But – as we now discuss in depth in the revised version of our manuscript (see specifically new Sec. C of Supplementary Note 2) the physics of DL in the discrete-time lattice setup goes well beyond the well-known Dunlap-Kenkre model and displays a richer physics that is so far unexplored. In particular, in the high-field driving regime the usual dynamics of ac-driven continuous-time lattices predicts band flattening and

frozen dynamics: in fact, the width of the quasi-energy band scales by the Bessel function factor $J_0(\Delta\phi)$, which shrinks to zero for very large values of the modulation amplitude $\Delta\phi$. Very surprisingly, the quasi-energy band flattening is not observed in discrete-time ac-driven systems as a result of the true discreteness of time dynamics: therefore, in the ultrastrong driving regime one observes **dynamical delocalization**, rather than DL. See specifically Fig. S2 in the Supplementary Material and our reply to Comment 6 of Reviewer 2 for more details, including Fig. R7. Therefore, we disagree with the Referee's view that DL realized in our discrete-time photonic quantum setup should be regarded as a mere extension of previous works on BOs. DL in ac-driven discrete-time systems can disclose **new physics**, which has been **largely unexplored so far in any experiment**.

(2) The use of the mesh lattice setup for the observation of DL localization provides some major advancements in the field of coherent light manipulation, as acknowledged by the other two Referees as well. With our setup we can demonstrate the strong robustness against noise of **high-order DL** regimes, which could not be reached with other setups (such as in geometrically-bent optical waveguide lattices). Also, the **temporal cloaking scheme** suggested and demonstrated in our experiment and based on high-/low-order DL switching regimes shows the rather unique property of being **reconfigurable**, enabling to dynamically reshape the cloaking region. The ability of tailoring and reshaping in a simple way the cloaking region is clearly important result that could be of potential relevance in applications. The detailed comparisons between DL and BOs in terms of cloaking are also provided in the reply to Comment 3 below. Therefore, we believe that our experimental results and underlying theoretical analysis provide indeed major advances and, as such, worth for consideration in Nature Communications.

(3) From an experimental perspective, the realization of DL is **more precise** and hence **more difficult** than that of BOs, not a merely transplant from dc to ac field driving. Under a dc electric field, one can observe BOs for arbitrary field amplitude because the uniform sweep across the entire Brillouin zone exists for an arbitrary-amplitude dc driving. However, under the ac driving, the collapse of the quasi-energy band structure occurs only for a series of particular driving amplitudes, i.e. the roots of the 0th-order Bessel function. In experiment, to satisfy this DL's triggering condition, one needs to carefully control the amplitude of the sinusoidally-varying phase modulation. However, the BOs of the wave packet could be observed under any linearly-varying phase modulation, i.e. the arbitrary dc electric field. We have briefly discussed the challenge of realizing DL in our experiment

by using the effective ac electric field. Please refer to the last two sentences of paragraph 1 in page 16 in the Methods section.

[Comment 2]

The dynamic localization has been explored in various platforms in experiments, and the current one does not show significant advance from the fundamental physical point of view. Of course, the high-order dynamic localization is novel. Nevertheless, such extension to realize the higher-order DL with the high-order zero solutions of Bessel function is well known, so I do not think that this novelty can guarantee the publication of the paper in NC.

Reply: (1) We respectfully disagree with the Referee's criticism. We fully agree that DL has been previously observed in different physical systems, and that observation of high-order DL itself could be considered as an incremental result. However, our experiments provide new major advances that are not acknowledged by the Referee. First of all, the unique characteristics and benefits of high-order DLs have not been demonstrated so far. We demonstrated for the first time that high-order DLs enable **stronger wave-packet localizations** than lower-order ones. As we also emphasized in our paper and demonstrated in the experiment, while in a clean system localization can be already achieved with low-order DLs, in disordered systems the **robustness** of DL is observed only for high-order DLs.

(2) As a second main point, as commented above the physics of DL in the **discrete-time** lattice setup goes well beyond the well-known Dunlap-Kenkre model of DL and displays a richer physics that is so far remained largely unexplored in any experiment. The realization of DL in a discrete-time ac-driven system could be thus of significant advance also from the fundamental physical point of view.

(3) As a third main achievement, the DL scheme in our setup is **fully reconfigurable**, i.e. one can dynamically shape the ac field on demand and thus reconfigure the wave packet paths. This functionality, which is desirable in applications such as **cloaking**, is clearly impossible to realize in other platforms, such as in geometrically-bent waveguide lattices, and never demonstrated so far to the best of our knowledge in any physical platform. Such achievements and the reconfigurability of the DL scheme demonstrated in our work clearly provide main advancements in coherent wave shaping, that deserves publication in Nature Communications.

In the revised manuscript, we have emphasized the novelty of our study based on the above discussions. Please refer to the last two sentences of the first paragraph and lines 3-13 of the last

paragraph in the introduction part.

[Comment 3]

The cloaking idea maybe attracting attentions at first glance. However, if one thinks it carefully, this so-called temporal cloaking is nothing but let the pulses avoid several sites in a one-dimensional mesh lattice when they are propagating versus the time and force them back near original positions later. Similar phenomena can be seen through different mechanisms in optics, such as a time-reversal operation (which by the way, I believe that the authors can do this operation too in the current setup without the use of DL). The studied temporal cloaking has no benefit to developments of the optical cloaking in real space.

Reply: We thank the Referee for this comment, which gives us the possibility to clarify the advantages and relevance of our cloaking scheme, showing its potential benefits in the development of optical cloaking devices. We fully agree with the Referee that the main idea of our cloaking scheme is to change the light path so as to avoid the scattering region, and then to force back the light to its original path. As correctly suggested by the Referee, the same effect could be achieved using other methods. Generally, the cloaking can be realized either by introducing a time-reversal operation onto the wave packet in two cascading regions or by exploiting the wave packet's self-imaging effects in a uniform region. For the time-reversal mechanism, the cloaking can be realized through the negative refraction of two split beams. Compared to the DL scheme, negative refraction requires negative refractive-index materials in real space^{1,2}, which is difficult to achieve. In synthetic dimensions, the negative refraction is usually realized through an out-of-phase shift of gauge potentials in two cascading regions³⁻⁶, but this method is not flexible and reconfigurable as the DL scheme we suggest and demonstrate. As a representative example of the self-imaging-based cloaking method, where the optical path is transiently deviated and then forced back toward its original path, is to exploit BOs, i.e. using a dc (rather than an ac) field. As a matter of fact, all methods based on time-reversal symmetry (as suggested by the Referee), high-/low-order DL (as suggested and demonstrated in our work) and BOs use an external control field to deviate the path of the optical beam away from the scattering region. So the following main question arises: is there any significant advantage in using the high-/low-order DL scheme, as compared to other scheme? Is there any benefit in the cloaking scheme suggested and demonstrated in our work, as compared to the other methods mentioned above? We can answer this

question claiming that our method offers for sure a great advantage because it is **dynamically reconfigurable**, enabling a very flexible design to shape the cloaking region. In our experiments we already showed that the cloak possesses flexible reconfigurability both in the cloaking window size and its opening time. Even more exciting, for a given fixed driving period the cloaking region can be dynamically reshaped. In fact, the shape of the cloaking region is basically given by the area enclosed by the paths followed by the two split beams, which can be estimated from the group velocities corresponding to two different bands

$$v_{g,\pm}(m) = \mp \cos(\beta) \cdot \sin[Q(m)] = \mp \cos(\beta) \cdot \sin[Q - \phi(m)]. \quad (\text{R8})$$

For a dc field α , leading to BOs, the momentum $Q(m) = Q + \alpha m$ changes linearly in time, so that the path followed by the beam exactly reproduces the shape of band structure. The main limits of using the BO approach (or other similar approaches such as time reversal) is that we cannot reshape the cloaking region: its shape always reproduces the shape of the energy dispersion curve. On the other hand, using an ac field we can control and dynamically reconfigure the path of the beam, for a fixed value of the driving period. By contrast, it is known that the path followed the quantum particle in the crystal under an ac field does not reproduce the shape of the dispersion curve and can be dynamically controlled, for example, by the initial phase of the ac field⁷. In particular, for a sinusoidal field $E_{eff}(m) = \omega \Delta\phi \sin(\omega m + \varphi)$, as in our experiment, by varying the phase φ of the field (from 0 to π) we can largely deform the beam path, i.e. reshaping the cloaking region. This is shown as an example in the attached movie (submitted as an attachment together with the revised manuscript), where we compare the beam paths under BOs (for a dc field $\alpha = \omega$) and under DL for increasing values of the phase φ . Parameter values are $\beta = \pi/3$, $\Delta\phi = 2.4$, $\omega = \pi/60$. Clearly, the movie shows that, at a fixed value of the duration time (defined by the period $M = 2\pi/\omega$), the beam path in the DL regime can be largely tailored by changing the phase of the sinusoidal field.

For a non-sinusoidal ac field, an even more flexible cloaking region reshaping could be realized. As an illustrative example, in Fig. R3 we show how the paths of wave packets (and thus the cloaking region) can be shaped by changing the ac field $\Delta\phi(m)$. The only requirement to observe self-imaging after one oscillation cycle is provided by the DL condition

$$\Delta n_{\pm} = \int_0^M v_{g,\pm}(m) dm = \mp \int_0^M \cos(\beta) \cdot \sin[Q - \phi(m)] dm = 0 \quad (\text{R9})$$

which reduces to the usual condition of $J_0(\Delta\phi)=0$ for the sinusoidal field, as demonstrated by our work.

Panel (a) reproduces our experiment, where we used a sinusoidal field at first-order DL point, i.e. $\Delta\phi(m) = \Delta\phi\cos(\omega m + \varphi)$ with $\Delta\phi = 2.4$, $\varphi = \pi/2$, $\omega = \pi/60$, $\beta = \pi/3$. Panel (b) corresponds to a modified ac field shaped as $\Delta\phi(m) = \Delta\phi_1\cos(\omega m + \varphi) + \Delta\phi_2\sin^3(\omega m + \varphi)$ with $\Delta\phi_1 = 1$, $\Delta\phi_2 = 3.1$, $\varphi = 2.5$, $\omega = \pi/60$, $\beta = \pi/3$.

These are just two illustrative examples; however they clearly demonstrate how the cloaking region can be reconfigured and dynamically shaped by simply acting on the signal voltage driving the phase modulators in the fiber loops. Our experiment thus proves the supremacy of the cloaking scheme based on the DL regime over other schemes, such as those based on BOs as suggested by the Referee. Dynamical reshaping of the cloaking region enabled by the high/low reconfigurable DL switching scheme is a powerful functionality which could be of major relevance in cloaking applications. Therefore, we fully believe that future developments of the reconfigurable optical cloaking in real fiber-based devices could greatly benefit from the scheme suggested and demonstrated in our experiment.

We have added the discussion about the cloak shape engineering in the revised manuscript. Please refer to the second paragraph of the discussion part in the main text, Supplementary Note 9, Refs. 34, 51-56 and also attached movie.

Related references in the above discussion:

1. Pendry, J. B. Negative refraction makes a perfect lens. *Phys. Rev. Lett.* **85**, 3966-3969 (2000).
2. Fang, N., Lee, H., Sun, C. & Zhang, X. Sub-diffraction-limited optical imaging with a silver superlens. *Science* **308**, 534-537 (2005).
3. Qin, C. *et al.* Spectrum control through discrete frequency diffraction in the presence of photonic gauge potentials. *Phys. Rev. Lett.* **120**, 133901 (2018).
4. Fang, K. & Fan, S. Controlling the flow of light using the inhomogeneous effective gauge field that emerges from dynamic modulation. *Phys. Rev. Lett.* **111**, 203901 (2013).
5. Yuan, L., Xiao, M. & Fan, S. Time reversal of a wave packet with temporal modulation of gauge potential. *Phys. Rev. B* **94**, 140303 (2016).
6. Qin, C., Wang, B. & Lu, P. Frequency diffraction management through arbitrary engineering of photonic band structures. *Opt. Express* **26**, 25721-25735 (2018).
7. Longhi, S. *et al.* Semiclassical motion of a multiband Bloch particle in a time-dependent field: Optical visualization. *Phys. Rev. B* **74**, 155116 (2006).

REVIEWER COMMENTS

Reviewer #2 (Remarks to the Author):

The authors have accurately addressed all comments and remarks, their response letter looks convincing. I therefore recommend the publication of the revised manuscript in Nature Communications.

Reviewer #3 (Remarks to the Author):

I have read the revised manuscript and the response. I find that the quality of the paper has been greatly improved after the authors considered the comments from all reviewers. In particular, I appreciate that the authors provided detailed responses to convince me that their work meets the standards for publication in Nature Communications. Moreover, the added clarity in explaining the novelty and major advances of their experimental results makes the revised manuscript qualified for the suggestion of acceptance. I therefore would be happy to suggest the publication of this revised manuscript in Nature Communications, with minor and optional follow-up comments:

1. I can see the importance of higher-order DL from the revised manuscript and also the response. I am wondering, since the higher-order DL point is more sensitive to parameters, maybe the use of lower-order DL is more stable in real applications. Can the authors discuss it a little bit more?
2. Since the temporal cloaking is an important claim in the paper, I am wondering if the authors can outlook any future work that can combine the temporal and spatial cloaking?

Response Letter to Manuscript ID: NCOMMS-22-10301A

High-order dynamic localization and tunable temporal cloaking in ac-electric-field driven synthetic lattices

by Shulin Wang, Chengzhi Qin, Weiwei Liu, Bing Wang, Feng Zhou, Han Ye, Lange Zhao, Jianji Dong, Xinliang Zhang, Stefano Longhi and Peixiang Lu

Reviewer 2:

Comments:

The authors have accurately addressed all comments and remarks, their response letter looks convincing. I therefore recommend the publication of the revised manuscript in Nature Communications.

Reply: We thank the Reviewer for his/her time and efforts spent to re-review our revised manuscript. We are very happy that the Referee finds our revisions, in reply to his/her remarks and comments, very accurate and convincing, endorsing publication of our revised manuscript in its present form.

Reviewer 3:

Comments:

I have read the revised manuscript and the response. I find that the quality of the paper has been greatly improved after the authors considered the comments from all reviewers. In particular, I appreciate that the authors provided detailed responses to convince me that their work meets the standards for publication in Nature Communications. Moreover, the added clarity in explaining the novelty and major advances of their experimental results makes the revised manuscript qualified for the suggestion of acceptance.

Reply: We thank the Reviewer for his/her time and efforts spent to re-review our revised manuscript. We are very glad reading that, after our revisions, the paper has been greatly improved and the Referee is now convinced that the manuscript meets the standards for publication in Nature Communications, fully acknowledging that that the novelty and major advances of our experimental results make the revised manuscript worth to be accepted for publication.

I therefore would be happy to suggest the publication of this revised manuscript in Nature Communications, with minor and optional follow-up comments:

Reply: We are very glad that the Reviewer endorses publication of our revised manuscript in Nature Communications, after considering some minor and optional comments.

[Comment 1]

I can see the importance of higher-order DL from the revised manuscript and also the response. I am

wondering, since the higher-order DL point is more sensitive to parameters, maybe the use of lower-order DL is more stable in real applications. Can the authors discuss it a little bit more?

Reply: We thank the Referee for suggesting this interesting follow-up comment. As we discussed in our revised version of the manuscript, and specifically with reference to Fig. S2 of the Supplementary material, in our discrete-time synthetic lattice system we have three different operational regions of dynamic localization (DL), depending on the order of resonance: (i) the low-order DL regime, corresponding to the first and second-order DL resonances; (ii) the moderate-to-high-order DL regime, corresponding to higher-order resonances up to, say, sixth order; (iii) the very high DL regime, corresponding to resonances above, say, ninth-order resonance. For real applications, where noise and/or imperfections in the system are unavoidable, the most stable operational regime is surely the moderate-to-high-order DL [regime (ii)]: here the localization condition is very tight and more robust against noise than low-order DL regime (see Supplementary Notes 3 and 6). However, one should avoid pushing the DL resonance condition to very high orders [regime (iii)]: in fact, in this case the discrete nature of the dynamics prevents collapse of the quasi-energy band, and thus localization, even in the absence of any external noise (see Fig. S2, bottom right panel, and Fig. R7 in our previous response to the Referee).

In the revised manuscript, we added a brief discussion the last two sentences, paragraph 2, page 15. In the revised Supplementary Note 2, this point has been clarified in the last paragraph in Sec. C.

[Comment 2]

Since the temporal cloaking is an important claim in the paper, I am wondering if the authors can outlook any future work that can combine the temporal and spatial cloaking?

Reply: We thank the Referee for this interesting comment about an outlook of possible future works that could combine temporal and spatial cloaking. In our current setup, the synthetic lattice is fully in the temporal domain and so the cloaking is limited to regions in one dimension (1D) and for temporal waveforms. Future works could extend the cloaking idea to regions in higher (e.g. 2D or 3D) synthetic dimensions. For example, with an adapted version of our current setup one could realize a 2D synthetic lattice fully in time domain^{1,2}: in this case one could design a 2D temporal cloak exploiting the DL-based principle demonstrated in our work in the 1D case.

Combining the temporal and spatial cloaking, i.e. realizing so-called spatio-temporal cloaking,

remains challenging, mainly because a space-time cloak involves the construction of a metamaterial that mimics propagation in a medium with a velocity $v(x, t)^{3,4}$, function of both space x and time t . However, thinking at physical space x as an additional synthetic dimension, like a time-bin as in our setup or in a synthetic frequency lattice, a high-dimensional cloak device might be envisaged.

In the revised manuscript, we outlook future works on the cloaking in lines 1-5, paragraph 1, page 15. The relevant references have also been added into the main text, see Refs. 57 and 58.

Related references in the above discussions:

1. Muniz, A. L. M. *et al.* 2D solitons in PT-symmetric photonic lattices. *Phys. Rev. Lett.* **123**, 253903 (2019).
2. Muniz, A. L. M., Wimmer, M., Bisianov, A., Morandotti, R. & Peschel, U. Collapse on the line – how synthetic dimensions influence nonlinear effects. *Sci. Rep.* **9**, 9518 (2019).
3. McCall, M. *et al.* Roadmap on transformation optics. *J. Opt.* **20**, 063001 (2018).
4. McCall, M. W., Favaro, A., Kinsler, P. & Boardman, A. A spacetime cloak, or a history editor. *J. Opt.* **13**, 024003 (2010).

REVIEWERS' COMMENTS

Reviewer #3 (Remarks to the Author):

I have read the response and the revised manuscript and I am happy to see that the authors have considered all my suggestions and discussed necessary concerns in the revised manuscript. At this stage, I want to recommend the publication in NC.

Response Letter to Manuscript ID: NCOMMS-22-10301B

High-order dynamic localization and tunable temporal cloaking in ac-electric-field driven synthetic lattices

by Shulin Wang, Chengzhi Qin, Weiwei Liu, Bing Wang, Feng Zhou, Han Ye, Lange Zhao, Jianji Dong, Xinliang Zhang, Stefano Longhi and Peixiang Lu

Reviewer 3:

Comments:

I have read the response and the revised manuscript and I am happy to see that the authors have considered all my suggestions and discussed necessary concerns in the revised manuscript. At this stage, I want to recommend the publication in NC.

Reply: We thank the Reviewer for his/her time and efforts spent to re-review our revised manuscript. We are very happy that the Referee finds our revisions, in reply to his/her remarks and comments, very accurate and convincing, endorsing publication of our revised manuscript in its present form.